# An investigation of the cognitive and neural correlates of semantic memory search related to creative ability

Marcela Ovando-Tellez [1✉], Mathias Benedek [2], Yoed N. Kenett [3], Thomas Hills[4], Sarah Bouanane[1], Matthieu Bernard [1], Joan Belo[1], Theophile Bieth[1,5] & Emmanuelle Volle [1✉]

Creative ideas likely result from searching and combining semantic memory knowledge, yet the mechanisms acting on memory to yield creative ideas remain unclear. Here, we identified the neurocognitive correlates of semantic search components related to creative abilities. We designed an associative fluency task based on polysemous words and distinguished two search components related to clustering and switching between the different meanings of the polysemous words. Clustering correlated with divergent thinking, while switching correlated with the ability to combine remote associates. Furthermore, switching correlated with semantic memory structure and executive abilities, and was predicted by connectivity between the default, control, and salience neural networks. In contrast, clustering relied on interactions between control, salience, and attentional neural networks. Our results suggest that switching captures interactions between memory structure and control processes guiding the search whereas clustering may capture attentional controlled processes for persistent search, and that alternations between exploratory search and focused attention support creativity.

[1] Sorbonne University, FrontLab at Paris Brain Institute (ICM), INSERM, CNRS, 75013 Paris, France. [2] Institute of Psychology, University of Graz, Graz, Austria. [3] Faculty of Industrial Engineering and Management, Technion—Israel Institute of Technology, Haifa 3200003, Israel. [4] Department of Psychology, University of Warwick, University Road, Coventry CV4 7AL, UK. [5] Neurology Department, Pitié-Salpêtrière hospital, AP-HP, F-75013 Paris, France. ✉email: marcela.ovandot@gmail.com; emmavolle@gmail.com

How creative ideas arise in our mind and in our brain is a key unresolved question. Ideas do not come from nowhere: It is commonly assumed that they result from searching, reorganizing, and combining the knowledge that is stored in semantic memory[1–9]. But specific mechanisms acting on memory to yield creative ideas are not well understood. Therefore, this study aims to identify processes of semantic memory search that support higher creative thinking, and explore the brain mechanisms supporting these processes.

Semantic memory search depends on the organization of semantic associations stored in memory that drives associative, spontaneous retrieval, and on controlled processes that navigate these retrieval processes based on the context and task demands. The role of semantic memory in creativity has been intensively examined across several lines of research[2,3,7,10–13]. The strength of semantic associations stored in memory, which determines their spontaneous activation by a given context, has been put forward as a core component in the creative process as early as in the 1960s[9]. Although associative thinking has proved challenging to measure empirically, studies have linked it to creativity using divergent thinking (generating different new and effective ideas) but also convergent thinking (finding solutions to problems by combining information in novel ways) tasks[3,11,14–19].

The organization of semantic associations, or semantic memory structure, can be studied as semantic memory networks (SemNets) via computational network science methods[20–22]. Network science is based on mathematical graph theory, and provides quantitative means to represent complex systems, such as semantic memory, as networks. In a semantic memory network, nodes represent concepts, and edges represent the relationship between these concepts based on similarity measures[22,23]. SemNet studies have explored how creative abilities are related to the semantic memory structure[7]. In this stream of research, the SemNets of higher creative individuals have been shown to be more connected, less spread out, and less segregated than those of lower creative individuals[6,7,24–27]. Such a SemNet structure has been linked to more creative search behavior[28]. Moreover, the link between SemNet structure and verbal creativity seems mediated by associative abilities[26].

Memory search is not only determined by the semantic memory structure, but also involves controlled processes that allow efficient retrieval and navigation within it[23,29,30]. Neuro-cognitive research on semantic memory identified a semantic control component supporting executive mechanisms that allows individuals to selectively retrieve, manipulate, and select meaningful information depending on context demands[31–37]. Controlled retrieval processes have been consistently related to creative abilities[3,38–41], as have been diverse executive abilities including working memory or inhibition of obvious and common responses[2,42–44].

Hence, both associative and controlled memory processes are thought to be jointly involved in creativity[2,10,14,45]. Then, when we search for ideas, what makes this search creative? What memory search components are related to higher creativity and how? Can we identify these components? One account arguing for the separability of these components was proposed in ref. [46]. They used verbal fluency tasks that typically assess semantic memory search[47–57], and characterized two components that interact during these tasks, so-called clustering and switching[46,51,58]. For example, in the category-fluency task, participants are asked to retrieve as many members of a category (e.g., animals) as possible in a limited amount of time (typically 1 or 2 min). During clustering, individuals generate words that belong to the same subcategory (e.g., birds). During switching, individuals jump to a different subcategory (i.e., from birds to amphibians). Troyer et al.[46] suggested that clustering reflects automatic semantic retrieval while switching involves executive processes. Although this hypothesis is coherent with cognitive models that discriminate two interacting types of cognitive processes (automatic versus controlled)[30,33,59–61], the alignment of clustering—switching with associative—executive components, respectively, is not established[31,52,54,62,63].

Hills et al.[49,50,64] further extended the separability of clustering and switching in the context of an exploitation–exploration trade-off, showing that memory search may involve a trade-off similar to the one found in spatial optimal foraging: switching to a different cluster occurs when the retrieval rate within the current cluster falls below a threshold[50]. By analogy, they proposed a cross-domain alignment of clustering–switching trade-off with local-global perception (e.g., perceiving the details vs. the global picture) or focal-diffuse attention (e.g., focusing to a specific information vs. widening the breadth of attention)[49]. Although alternative views exist (e.g., refs. [47,65,66]), the framework of exploration and exploitation in semantic foraging offers a useful quantitative method to characterize the responses in diverse generation tasks relevant to creativity. For instance, a few studies have separated exploitation/exploration processes operationalized in a visuospatial creativity task[67] or clustering and switching behavior in a divergent thinking task[68–70], suggesting that they reflect separable processes supporting creative ideation.

The above accounts argue for separable semantic search processes, but little is known about their individual contributions to creative cognition[71]. Broad retrieval ability, the ability to fluently retrieve semantic information from long-term memory, has been reliably associated with creative performance[2,39–41], suggesting that semantic search plays an important role for creative thinking in general. Switching during semantic memory search has been shown to be related to executive processes[72,73] and may capture control processes which are particularly critical for the search of creative ideas[68,70]. This is further suggested by correlations found between creative thinking and diverse executive functions[38–41,45]. In addition, creative people have been shown to be able to switch between alternative meanings and consider things from different perspectives, allowing them to re-structure their mental representations[74–79]. Overall, the clustering and switching components of memory search may both play a role in creativity. Their contribution may vary according to the cognitive processes presumably supporting each component, i.e., memory structure or control processes, respectively. The role of clustering and switching in creativity may also depend on the relative importance of controlled and spontaneous processes in the creativity tasks that are used.

Assessing the neural correlates of clustering and switching offers another perspective for understanding the role of semantic structure and cognitive control in memory and creative search. Functional neuroimaging studies[80–82] and lesion studies[83,84] have shown that memory search during category-fluency tasks relies on the coordinated activity of several brain areas including the left inferior frontal gyrus, medial prefrontal areas and premotor regions, cingulate cortex, insula, middle frontal gyrus, and anterior and posterior temporal regions. The specific role of these regions in memory search is not fully understood. Troyer et al.[46] proposed that clustering depends on the temporal lobe memory regions, while switching reflects executive frontal lobe processes. However, empirical support for such dissociation is weak and existing studies show a more complex picture[46,54,85,86]. Besides, functional connectivity studies on creativity have shown that processes of idea generation are supported by the interaction of several large-scale brain networks[15,87–89] including the default mode network (DMN), executive control network (ECN), and salience network, rather than individual regions. The DMN is thought to support associative thinking and spontaneous retrieval

allowing for the generation of candidate ideas while the ECN participates in controlled retrieval, constraining the search, and monitoring the responses for appropriateness. Yet, how these networks relate to memory search processes is unclear[18,19,90]. Assessing functional brain connectivity associated with individual differences in clustering and switching may provide a useful approach revealing the neural correlates that underlie distinct memory search components.

To explore the neurocognitive components of semantic search that relate to creative abilities, we designed an associative fluency task in which a free association search is initiated from an ambiguous polysemous word (PolyFT). Ambiguous, polysemous words are characterized by having different meanings, thus more than one concept may be activated in memory for the same word (e.g., for the word bank, there are at least two meanings: river bank and bank account[74]. The advantage of this task is that ambiguous words are helpful in investigating the clustering and switching framework, because they allow a clear separation of the clusters based on the different meanings. Such a task enables us to assess clustering and switching based on the number of responses within and between the different meanings of the ambiguous cue word. Cluster definition has indeed been a limitation in previous methods using category-fluency tasks, as subcategories are difficult to define or have blurred borders (e.g., by context (zoo) or by taxonomy (birds))[54]. Although a few computational approaches have defined clusters based on semantic similarity measures, they remain dependent on normative data or text corpora[68,91,92]. In addition, ambiguous words are relevant to the exploration of creativity for several reasons. First, creative people have been shown to be better able to activate the representation of multiple aspects of potentially incongruous information, including competitive lexico-semantic meanings[74] or figures[75,79]. Second, ambiguous words have proven useful in isolating control demands in previous studies on semantic memory[86,93,94].

The PolyFT enabled us to assess individual differences in clustering and switching behavior, which are theorized to be key components of memory search related to creativity[68–70]. We examined how these two components relate to creative abilities, individual differences in semantic memory structure using SemNets[24–27], and executive abilities[84]. Finally, using connectome-based predictive modeling (CPM) approach[88], we explored the functional connectivity patterns that predict clustering- and switching-related components. This approach allowed us to address four hypotheses: First, we expected that both clustering- and switching-related components (as assessed with PolyFT) would correlate with creativity task performance. Such a finding would extend the seminal work from ref. [46] to creativity and show that creative thinking involves similar cognitive processes associated with semantic memory search. Second, we hypothesized that switching-related measures would correlate with executive abilities[46,72,73]. Third, because previous research demonstrated that producing a chain of related words (associates) involves a spontaneous and unconstrained mode of retrieval with little executive demands[16,18] we expected the clustering-related component of the PolyFT would be related to semantic memory structure as captured by SemNets and more limited executive control. The findings arising from the second and third hypotheses would clarify how the processes framed in different constructs (memory search components, semantic structure, executive functions) relate or differ from each other, and how they correlate with creative abilities. Fourth, we expect that clustering and switching are associated with discriminable brain activation in terms of connectivity patterns between the ECN, DMN, and salience networks.

Our findings allowed us to characterize two semantic memory search components related to clustering and switching in the context of semantic memory structure, executive abilities, and brain connectivity patterns. Both components correlated to creative abilities, but differently. Based on the cognitive and neural patterns of each component, we propose that the switching component captures interactions between memory structure and control processes guiding search that support the ability to combine remote associates, whereas the clustering component captures controlled processes related to the persistent search that support divergent thinking. These results help to better understand the role of semantic memory search in creative cognition.

## Results

**Principal component analysis of the ambiguous word-fluency task (PolyFT) measures.** In the PolyFT task, participants were required to name all the words that they could think of as associated with ambiguous, polysemous cue words (i.e., free word associations) presented successively. We adapted the method in ref. [46] to quantify five different measures related to clustering and switching with respect to the different meanings of ambiguous cue words (Supplementary Table 1): fluency, rank of the first switch between meanings, number of different meanings, number of switches between meanings, and bigger cluster size within the same meaning (see "Methods"). To reduce the set of dimensions of the PolyFT task and because several measured variables were strongly correlated (see the correlation matrix between the five measured variables in Supplementary Table 2), we conducted a principal component analysis (PCA) of the five PolyFT measures. We identified two factors with eigenvalues higher than one that together explained 83% of the variance. Table 1 shows the component's loadings after oblimin rotation. The first one captures fluency, biggest cluster size, and rank of the first switch. Hence, this first component likely reflects clustering of responses within semantic meanings. The second component captures the number of different meanings and transitions between them (i.e., number of switches), thus likely reflecting switching-related processes[51]. The correlation between the two components was not significant ($r_s = -0.153$, $P = 0.160$) supporting their discriminant validity.

Overall, these results indicate that although nuanced aspects of semantic search were separately assessed, they essentially reflect two distinct factors that can be named clustering and switching. We proceeded using these component scores for further analyses.

**Relationship between PolyFT clustering and switching and other behavioral measures.** We used Spearman correlations to explore the relationships between the identified principal

**Table 1 Principal component analysis (PCA).**

| PolyFT scores | Components | |
|---|---|---|
| | Clustering | Switching |
| Fluency | **0.983** | 0.339 |
| Rank of the first switch | **0.668** | −0.409 |
| Number of different meanings | −0.093 | **0.827** |
| Number of switches | 0.085 | **0.937** |
| Biggest cluster size | **0.892** | −0.197 |
| Eigenvalues | 2.599 | 1.549 |
| % of variance | 51.978 | 30.982 |
| Cumulative % | 51.978 | 82.960 |

The PCA analysis performed on the five PolyFT measures, showing the loadings of the variables on the two principal components extracted from the data of the 86 participants. For each component, we also report the percentage of variance explained. Absolute values of regression coefficients higher than 0.5 were considered important (bold font) in defining the principal components.

**Table 2 Behavioral measurements.**

| Task | Parameter | Measurement |
|---|---|---|
| **Creativity tasks** | | |
| Alternative uses task (AUT) | AUT-fluency | Fluency—number of responses |
| | AUT-uniqueness | Uniqueness of responses |
| | AUT ratings | External ratings to the top-2 responses |
| | AUT-commonness | Overall frequency of responses |
| Combination of associates task (CAT) | CAT-CR | Accuracy in performance |
| | CAT-index | Performance in distant trials relative to close trials |
| | CAT-eureka | Insight report in correct trials |
| **Semantic Network (SemNet) metrics** | | |
| Weighted undirected networks (WUN) | WUN Eff | Global efficiency |
| | WUN CC | Clustering coefficient |
| | WUN Q | Modularity |
| Unweighted undirected networks (UUN) | UUN Eff | Global efficiency |
| | UUN CC | Clustering coefficient |
| | UUN Q | Modularity |
| **Executive tests** | | |
| Digit span test | Forward-span | Working memory verbal span in direct order |
| | Backward-span | Working memory verbal span in reversed order |
| Fluency tasks | Category-fluency | Broad retrieval abilities in a semantic category |
| | Letter-fluency | Broad retrieval abilities with a phonological constrain |
| Trail-making test (TMT) | TMT-shifting | Set-shifting assessed by the difference in the time taken to perform part B and part A |
| Stroop test | Stroop-interference | Cognitive inhibition assessed by the difference in time to complete the third and (ink naming with interference) and second (color naming) part |

A summary of the creativity tasks, SemNet metrics, and executive tests used in the analyses. We provide the acronym in parenthesis for each task when appropriate, the different parameters we measured and what they measure.

components of the PolyFT task and other behavioral measures (Table 2) of creative abilities, semantic memory structure via SemNet metric computation, and executive abilities. The descriptive statistics of creativity tasks, SemNet metrics and executive tests are reported in Supplementary Table 3. The correlations between PolyFT components, creativity tasks, SemNet metrics, and executive tests are reported uncorrected and corrected for multiple comparisons using the false discovery rate (FDR) in Fig. 1 and Supplementary Table 4.

**Relationship between PolyFT clustering and switching and creativity.** To assess creative abilities, we used two validated creativity tasks. Divergent thinking was assessed with the Alternative Uses Task (AUT)[95]. Convergent thinking was assessed with the combination of associates task (CAT)[9,15,96] (see "Methods").

We examined correlations between clustering and switching components of PolyFT and the two classic creativity tasks used (see Table 2). Spearman correlations analyses revealed significant correlations between the clustering component and AUT-fluency ($P < 0.001$), AUT-uniqueness ($P < 0.001$), and AUT-commonness ($P = 0.013$). The switching component was positively correlated with the accuracy in CAT task CAT-CR ($P = 0.032$). The other correlations between the PolyFT components and creativity scores were not significant ($P > 0.05$). The correlations between clustering and both AUT-fluency and AUT-uniqueness remained significant after FDR correction for multiple comparisons. These results indicate that individuals with higher clustering in the PolyFT task generate a higher number and more unique and infrequent ideas in the AUT, whereas individuals with higher switching in the PolyFT task are better able to combine remote elements in the CAT task (see Fig. 1 and Supplementary Table 4). Overall, the clustering component was more related to divergent

thinking (AUT) while the switching component was more related to convergent thinking (CAT).

**Relationship between PolyFT clustering and switching and individual semantic network metrics.** We developed a relatedness judgment task (RJT; see "Methods") to estimate the individual SemNets (Fig. 2a)[24–26]. Based on the RJT ratings, we built two types of individual SemNets whose properties have been related to creative abilities in previous studies (Fig. 2b)[24–27]. For each individual, we built the weighted undirected network (WUN) and the unweighted undirected network (UUN; see Supplementary Note 1). For each type of graph, we computed their network metrics[97–100] selected based on previous studies showing their relevance for creativity[24–27]. We estimated the global efficiency (Eff), clustering coefficient (CC), and modularity (Q) of the SemNets (see Table 2), representing how efficient is the structure of the network to allow the transition of information (Eff), the level of connectivity (CC) and segregation (Q) of the network, respectively. A more flexible SemNet has been related to higher Eff and CC, and lower Q[6,7,24–27].

We tested whether clustering and switching in PolyFT related to individual differences in the SemNet structure, using Spearman correlations (Fig. 1 and Supplementary Table 4). The analysis showed a significant correlation between the switching component and Eff in both the WUN ($P = 0.023$) and UUN networks ($P = 0.007$). In addition, the switching component correlated negatively with Q ($P = 0.023$), and positively with CC ($P = 0.019$) in the UUN network. The correlation between Eff in the UUN network and the switching component remained significant after FDR correction. These results suggest that individuals with higher switching during the PolyFT task had a more efficient SemNet.

**Relationship between PolyFT clustering and switching and executive functions.** We used validated neuropsychological tests

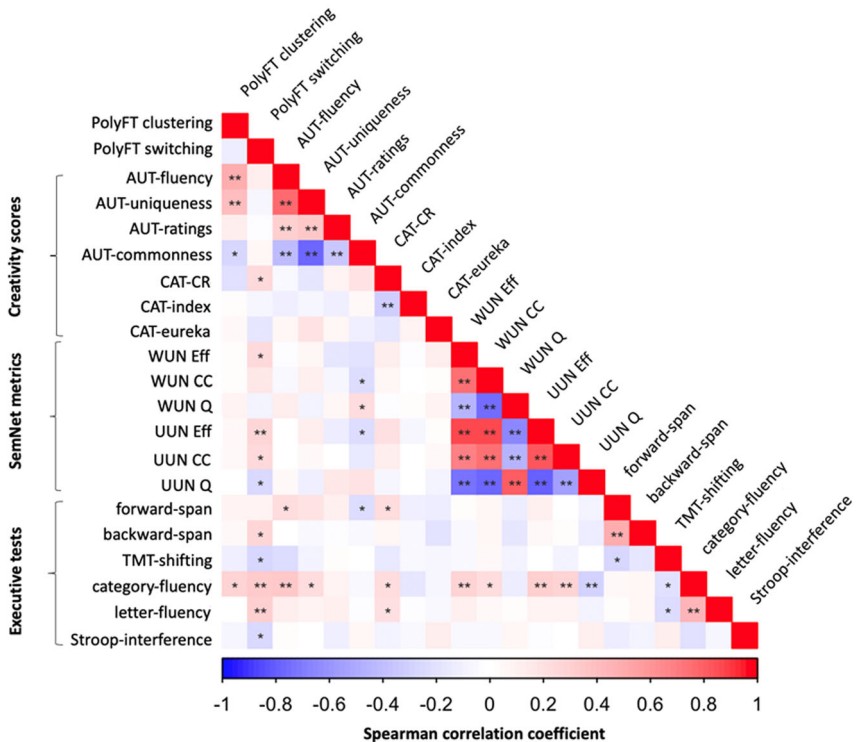

**Fig. 1 Correlation matrix of search components with other behavioral measures.** Spearman correlations between PolyFT clustering and switching components and SemNet metrics, creativity scores (AUT and CAT tasks) and executive function tests ($n = 86$). Cold to hot colors code the Spearman correlation coefficient ($r_s$). *$P < 0.05$; **$P < 0.01$. All correlations signaled with **($P < 0.01$) remained significant after FDR correction for multiple comparisons.

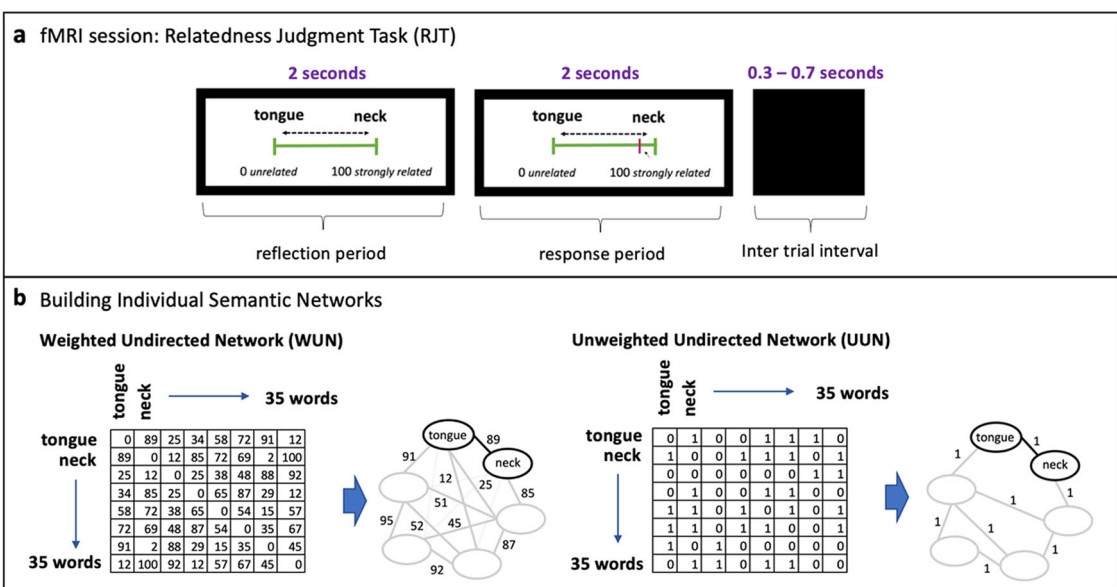

**Fig. 2 Building of individual semantic networks (SemNets) from the RJT. a** In each RJT trial, participants judged the relatedness of 595 word pairs. During the reflection period, participants thought about the relationships between the different pairs of words. In the response period, participants moved the cursor (in magenta) using a visual scale ranging from 0 (unrelated words) to 100 (strongly related words) to indicate the relatedness of the two words. A jittered inter-trial interval separated trials. **b** Using the ratings of all the RJT trials, we computed a 35 by 35 adjacency matrix for each participant with rows and columns representing the words of the task. For the weighted undirected networks (WUN), cell values correspond to the relatedness judgments given by the participant during the RJT. For the unweighted undirected network (UUN), cell values correspond to a binary value of 1 for judgments above or equal to 50, and 0 for judgments below 50. From these adjacency matrices, we built the WUN and UUN graphs and estimated their SemNet metrics[24,27].

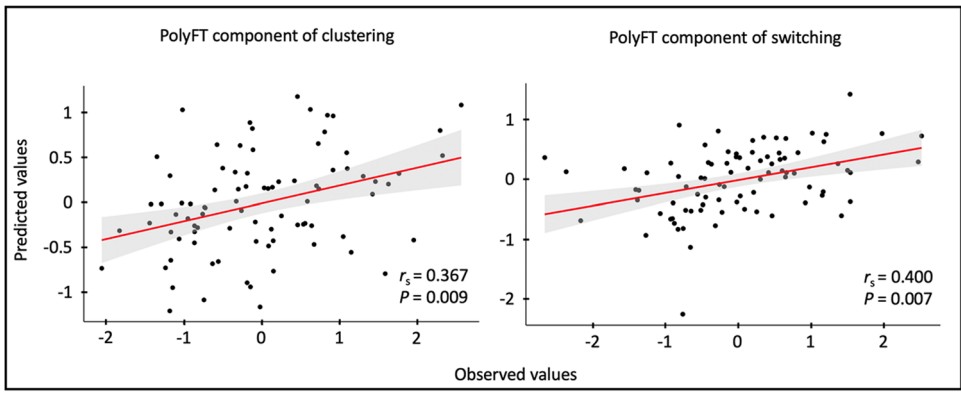

**Fig. 3 Predicted and observed PolyFT components.** The plots show the Spearman correlations between the predicted values (*y* axis) and observed values (*x* axis) of the clustering and switching components based on brain connectivity for the significant predictions (*n* = 86). The fitting regression line is shown in red, and the 95% confidence interval for the line is shown in gray shading. For each plot, we present the $r_s$ and the *P* values. The reported *P* values are based on permutation testing.

to assess executive processes (Table 2; see Supplementary Table 3 for descriptive statistics). We explored Spearman correlations between the PolyFT components and the executive function tests. The clustering component correlated positively to category fluency (*P* = 0.014). The switching component was positively correlated to backward-span (*P* = 0.032), category-fluency (*P* = 0.001) and letter-fluency (*P* = 0.008), and negatively correlated to TMT-shifting (*P* = 0.019) and Stroop-interference (*P* = 0.017). After FDR correction, the correlation of the switching component with category-fluency and letter-fluency remained significant (see Fig. 1 and Supplementary Table 4). While both components of the PolyFT related to retrieval abilities in the semantic fluency task, only switching related to executive abilities.

**Brain correlates of PolyFT clustering and switching: predictions from brain connectivity using CPM.** To identify the brain substrates of the clustering and switching components, we explored the functional connectivity patterns predicting these scores, using a CPM method with a leave-one-out cross-validation[27,88,101,102]. Based on the Schaefer brain atlas[103], we defined 200 regions of interest (ROIs) distributed into 17 functional subnetworks organized in eight main functional networks. We performed Pearson pairwise correlations of the blood oxygen level-dependent (BOLD) signal between all unique pairs of brain regions (i.e., ROIs). As a result, we obtained a 200 × 200 matrix for each participant corresponding to their functional connectivity network in which ROIs are the nodes and correlation coefficients the links. We followed the method described in ref. [102] regressing a cognitive component (i.e., clustering and switching components in separate CPM analyses) on brain networks with a leave-one-out cross-validation. First, we identified the links that significantly correlated with each of the PolyFT components (*P* < 0.01) either positively (the positive model network) or negatively (the negative model network) across participants (*N* − 1 participants). We then estimated the connectivity strength in these model networks for each participant by summing the functional connectivity values of the selected ROI pairs. We built a linear model with the resulting individual connectivity strength in the positive and negative model networks as predictors and the PolyFT component as the outcome. Finally, we applied the predictive linear model to the left-out participant and obtained a predictive value of the PolyFT component for each participant. We used Spearman correlations between the predicted and the observed values of the PolyFT components to test the validity of the prediction. The predictive power of the predictions was evaluated using permutation testing.

The results (Fig. 3) revealed significant correlations between the predicted and observed clustering component (*r* = 0.367, *P* < 0.001) and between the predicted and observed switching component (*r* = 0.400, *P* < 0.001). To test the robustness of the prediction, we analyzed the relation between model-predicted and observed components, using 1000 iteration permutation testing. Both predictions remained significant after the permutation testing (*P* = 0.009 for the clustering component; *P* = 0.007 for the switching component). These findings suggest that patterns of brain functional connectivity allow robust predictions of both clustering and switching components.

Finally, we characterized the brain regions and functional networks of the positive model networks (models in which higher connectivity was associated with higher scores) that account for a higher clustering and switching during the PolyFT task. (see Supplementary Note 2 and Supplementary Figs. 1 and 2 for the negative predictive networks).

The positive predictive network of clustering included 43 links with a whole-brain distribution (Fig. 4). The majority of the links (63% of all connections) connected brain regions belonging to the executive control networks (ECN; in particular the intraparietal sulcus) to diverse regions of other networks, including the salience, visual, and somatomotor networks. The model network also included links between regions of the dorsal attention network (DAN; in particular the superior parietal lobule) and regions of the visual and somatomotor networks, as well as between the temporal pole (Supplementary Fig. 3a) and several regions of the salience network (23% of all connections). No links were observed within the ECN or within the salience network. The nodes with the highest number of connections (node degree *k*; a total of nodes with *k* > 0 = 42) were localized in the left intraparietal sulcus region of the ECN (*k* = 26) and right superior parietal lobule in the DAN (*k* = 6). The intraparietal sulcus (ECN) had the most links with salience, visual, and somatomotor network regions.

The switching model network was composed of 259 links with a whole-brain distribution (Fig. 5). A higher number of links connected different brain regions belonging to the DMN to regions of the salience (in particular the frontal operculum), DAN, visual, and somatomotor networks, and to the lateral prefrontal and temporal areas of the ECN (56% of all connections). Links between the DMN and the temporal pole (limbic network) were also observed (Supplementary Fig. 3b). No links within DMN network were observed. The model network also included several connections between ECN regions and regions of the visual network as well as between the salience

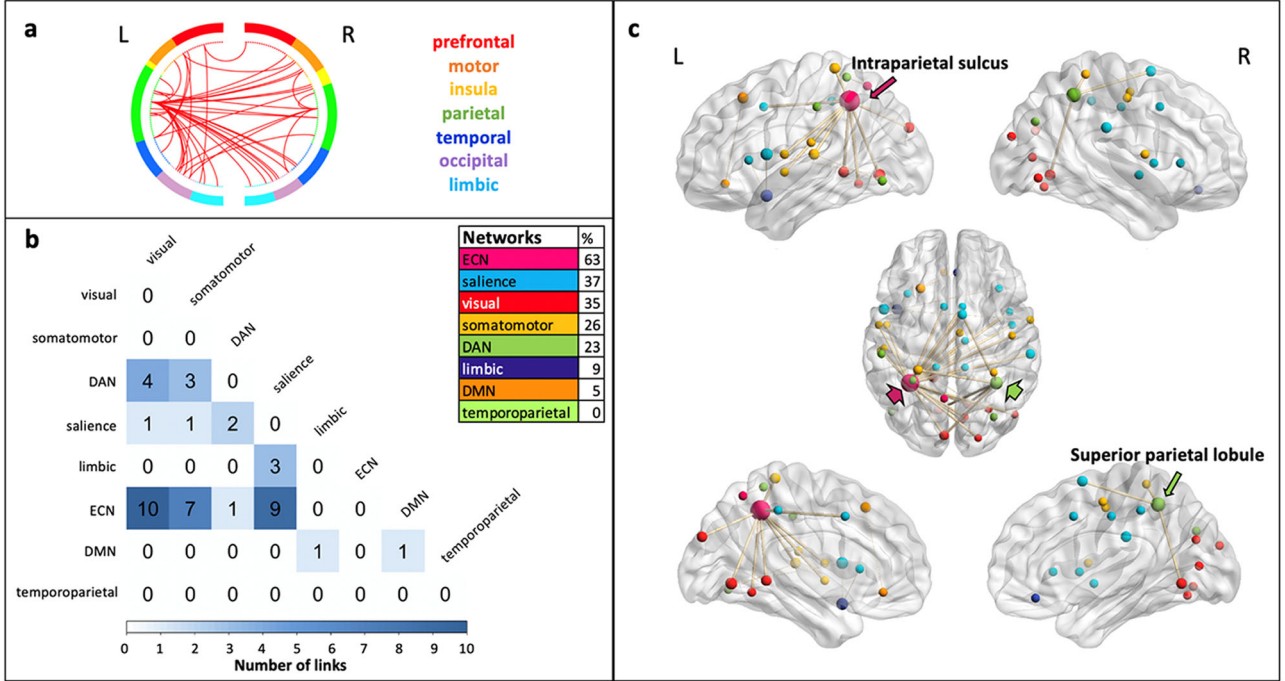

**Fig. 4 Functional anatomy of the CPM model predicting the PolyFT clustering.** The functional connectivity patterns of the positive model network predicting clustering are described. **a** The circular graph represents the distribution of links within and between brain regions in the left (L) and right (R) hemispheres. Brain regions are color-coded. The lines in red represent the links connecting the ROIs. **b** We examined the distribution of the links across intrinsic functional networks based on Schaefer's atlas[103]. The correlation matrix represents the number of links within the model network connecting seven different brain lobes (total of links = 43). A table with the percentage of links connecting nodes that belong to these functional networks is shown. This percentage considers all connections to the regions of a given functional network in relation to the total number of connections in the model network (i.e., 43 links). **c** The nodes (total of nodes = 42) and links of the model network are superimposed on a volume rendering of the brain. From top to bottom, a lateral, dorsal, and medial views of the left (L) and right (R) hemispheres are shown. The color of the nodes represents the functional network they belong to, using a color code presented in **b**. The size of the nodes is proportional to their degree. Thicker links are connecting the highest degree nodes to the rest of the brain. The color code arrows indicate the highest degree nodes and the brain region in which they are localized. For visualization purposes, nodes with degree $k = 0$ are not displayed. ECN executive control network, DMN default mode network, DAN dorsal attention network.

network and the visual and somatomotor networks. We also observed in particular intra-ECN links between its lateral prefrontal and temporal regions. The brain regions with the highest number of connections (node degree $k$; total of nodes with $k > 0 = 122$) were localized in the left orbital frontal cortex of the limbic network ($k = 31$), bilateral inferior parietal lobule of the DMN (left $k = 28$; right $k = 12$), the left retrosplenial region of the DMN ($k = 24$), bilateral extra-striate inferior regions of the visual network (left $k = 20$; right $k = 17$), left medial prefrontal cortex of the DMN ($k = 15$) and the right precentral region of the salience network ($k = 12$). The orbital frontal cortex of the limbic network had most links with regions of the DAN, salience, somatomotor, and visual networks. The retrosplenial region of the DMN connected mostly to salience and DAN regions. The extra-striate inferior regions of the visual network connected mostly to the DMN, ECN, and salience network regions. The medial prefrontal cortex in the DMN had main connections with the DAN, salience, and visual network regions. The prefrontal region of the salience network had several links with the DMN.

In summary, the main patterns of functional connectivity predicting clustering and switching components of PolyFT differed considerably. Higher clustering was predicted by higher connectivity between large-scale brain networks, in particular between ECN, salience, DAN, and visual networks, but, notably, the clustering model network hardly included any connectivity within these networks. Higher switching was predicted by a higher connectivity between the DMN, ECN, salience, DAN,

somatomotor, and visual networks; yet, contrary to the clustering model network, links between the ECN and the DMN played a more important role in the switching model network, and DMN regions were generally much more involved. The salience network is mostly related to the ECN in the clustering model but with the DMN in the switching model network. The temporal pole, a key memory region, connected to salience network regions in the clustering model network but to DMN regions in the switching model network (Supplementary Fig. 3). Only the switching model network involved intra-ECN connectivity. Both model networks involved the visual and salience networks.

**Internal validation: prediction of PolyFT clustering and switching from resting-state functional connectivity.** As a final step, we explored whether the predictive models we trained on the task-based functional connectivity data generalize to resting-state data. We performed an internal validation by predicting the PolyFT components from the participant's resting-state data. The Spearman correlations between the predicted value from the model applied to the resting-state data and the observed values showed significant predictions for clustering ($r_s = 0.484$, $P < 0.001$) and switching ($r_s = 0.214$, $P = 0.048$) components. These significant predictions suggest that clustering and switching abilities in the PolyFT task are reflected in intrinsic connectivity, and provide support to the CPM-based predictive model's robustness and its generalization to the functional connectivity during resting-state data.

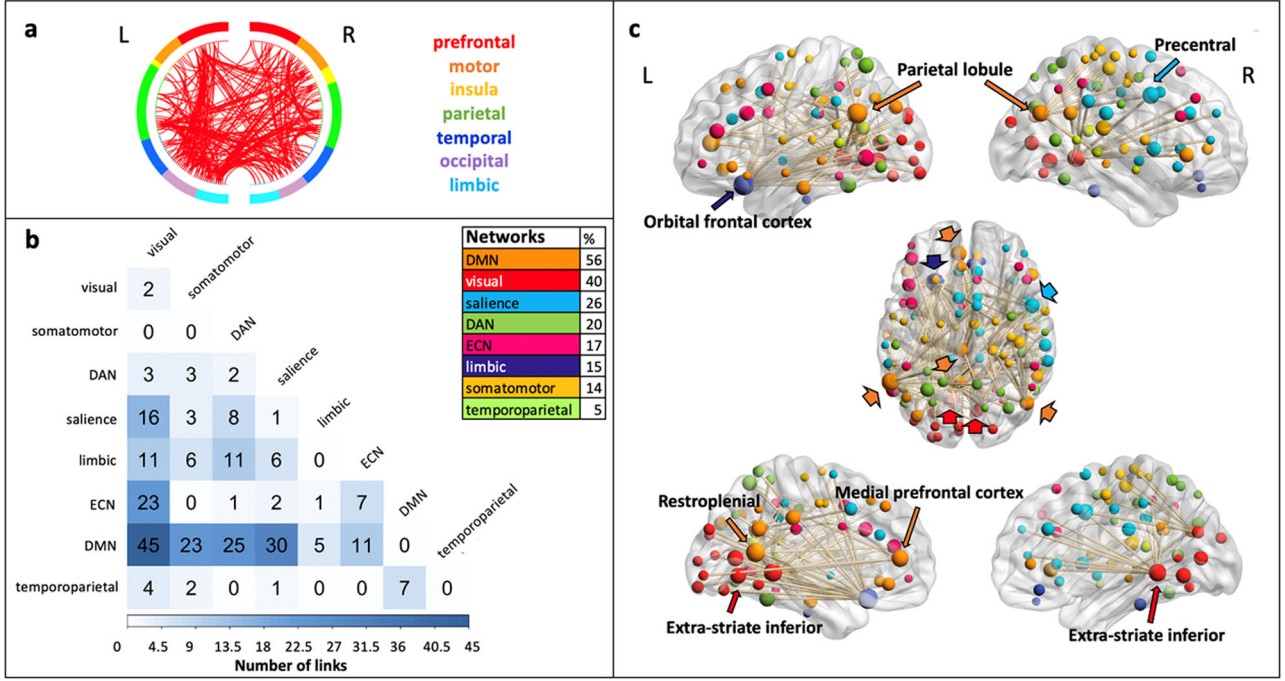

**Fig. 5 Functional anatomy of the CPM model predicting the switching PolyFT component.** The functional connectivity patterns of the positive model network predicting switching are described. **a** The circular graph represents the distribution of links within and between brain regions in the left (L) and right (R) hemispheres. Brain regions are color-coded. For visualization purposes, we used a nodal degree threshold of $k > 10$. The lines in red represent the links connecting the ROIs. **b** We examined the distribution of the links across intrinsic functional networks based on Schaefer's atlas[103]. The correlation matrix represents the number of links within the model network connecting within and between eight intrinsic brain networks (total of links = 259). At the upper right side, a table with the percentage of links connecting nodes that belong to these functional networks is shown. This percentage considers all connections to the regions of a given functional network in relation to the total number of connections in the model network (i.e., 259 links). **c** The nodes (total of nodes = 122) and links of the model network are superimposed on a volume rendering of the brain. From top to bottom, a lateral, dorsal, and medial views for the left and right hemispheres are shown. The color of the nodes represents the functional network they belong to, using a color code presented in **b**. The size of the nodes is proportional to their degree. Thicker links are connecting the highest degree nodes to the rest of the brain. The color code arrows indicate the highest degree nodes and the brain region in which they are localized. For visualization purposes, nodes with degree $k = 0$ are not displayed. ECN executive control network, DMN default mode network, DAN dorsal attention network.

## Discussion

This study investigated the neurocognitive correlates of memory search related to creative cognition. We developed a free association task based on ambiguous words, which was shown to capture two main components: a clustering component related to fluency and cluster size of responses, and a switching component related to the diversity and flexibility in these responses and their meanings. We found that clustering and switching components were differently related to divergent and convergent thinking and that the switching component, but not clustering, was associated with individual differences in semantic memory structure (measured by SemNet metrics) and executive control. The application of network science methods at the neural level further allowed us to identify differences in their respective predictive brain connectivity patterns consistent with the behavioral results.

Both clustering and switching components correlated with creativity performance. However, they differed in their relationships with divergent and convergent creativity tasks. The clustering component correlated with AUT measures of fluency and originality (number of total and unique ideas), suggesting that it captures processes involved in divergent thinking. This result converges with previous studies that related performance in various kinds of verbal fluency tasks to fluency and originality in divergent thinking tasks[2,3,14,18,24,40,41,68,104], showing that broad retrieval ability is consistently associated with creative ability[39,41]. It is important to mention that clustering did not correlate with a measure of AUT originality that is not confounded by fluency in

AUT (AUT ratings of top-creative responses), which may suggest that the observed correlations with AUT-uniqueness were driven by the fluency[105–107]. Effective exploitation of semantic clusters may thus drive fluent idea generation.

In contrast, we did not observe correlations between the switching component and divergent thinking scores in the AUT task, but with the ability to combine remote associates in the CAT task, a convergent thinking measure of creativity that requires converging on a single solution by considering unrelated semantic concepts. The link between flexible retrieval abilities and CAT suggests that combining unrelated semantic concepts in the CAT is supported by flexible switching between semantic clusters[7,9,13,17]. Importantly, the link between CAT performance and the switching component is consistent with previous findings showing that optimal search in the Remote Associates tests involves continuous switching between clusters of associates[108] (but see ref. [109]) and involves controlled processes[15,109–111]. Switching-related control processes may thus relate to an exploration function involved in solving CAT problems.

The relationships of the PolyFT components with SemNets metrics and executive function tests may help to better understand the underlying processes. As expected, the clustering and switching components differ in their relationships with semantic memory structure and executive control measures, suggesting that they may capture distinct semantic and/or executive processes involved in memory search (see Fig. 1). While both components correlated with category-fluency, only the switching

component related to executive abilities, including letter-fluency, and (although uncorrected for multiple comparisons) inhibition of automatic responses (Stroop-interference), cognitive flexibility (TMT-shifting), and working memory (backward-span). This finding is in line with previous research showing that switching between subcategories in fluency tasks relies on cognitive control as evidenced by correlations with frontal control functions[46,53,54,85]. Similarly, the processing of ambiguous words likely involves executive processes to retrieve multiple unrelated meanings of the words[86,93], manipulate and update the content of working memory, inhibit previously visited meanings that would not be profitable to revisit, and switch between semantic fields.

Based on previous work on associative fluency tasks[18,26] and on Troyer's hypothesis[51] we expected components of the PolyFT task, a free association task, to capture individual differences in the semantic memory organization explored via the SemNet properties. The switching component indeed correlated with SemNets efficiency (significant correlations with modularity and CC did not survive the FDR corrections), in agreement with a previous result[112], indicating that more efficient information spread in semantic memory is associated with more effective switching. Hence, self-paced switching, allowing to alternate between diverse meanings during the PolyFT task, is related to individual differences in semantic memory structure. Overall, switching in PolyFT seems to rely on both semantic memory structure and executive control processes, which together may support flexibility in memory search.

Contrary to our expectations, the clustering component did not correlate with SemNets metrics. Because the participants were asked to give unconstrained free associations during the PolyFT task, and based on Troyer's hypothesis, we expected clustering to capture spontaneous semantic associations from the underlying semantic memory network[18,26]. However, the absence of significant correlations with SemNets and the evidence for activity in the ECN predicting clustering questions this interpretation. One potential explanation is that clustering may involve additional control processes and is therefore not exclusively driven by differences in the organization of semantic memory. Staying within a cluster may require a focused, sustained, goal-directed attention to suppress interference proactively. This interpretation converges with the exploration-exploitation trade-off extended to semantic search[49,50], where switching to a different cluster occurs when the retrieval rate within the current one falls below a threshold. Hence, our clustering component may capture the ability to maximally exploit a given meaning, or the tendency to persist longer in a local/exploitation mode, as opposed to being a global/exploration mode completely free of executive control. Further studies directly testing the impact of a lack of executive control on clustering and switching are needed to clarify this point. We can also not exclude that clustering relates to a prototypical semantic memory structure rather than to an individual difference in SemNet structure.

Overall, the switching component relates to individual differences in the organization of semantic memory networks, executive function abilities, and the ability to combine remote associates measured with the CAT task. These results suggest that switching captures a self-paced flexible search behavior during the PolyFT task that may relies on both semantic memory structure and executive processes. This behavior may be particularly relevant during convergent thinking tasks such as the CAT, where exploration and switching between different types of relationships between the cue words and the solution word facilitate performance[108,109]. Conversely, the clustering component is related to semantic retrieval and fluent idea generation and may capture controlled processes to maximally exploit clusters and maximize the number of responses. Hence, effective clustering

involves more cognitive control than may have been previously expected.

This suggests that controlled processing is involved in both clustering and switching but in different ways. These are exemplified by the correspondence between the two mechanisms proposed for memory search (exploitation/exploration)[50]. That is, the clustering and switching components may reflect an exploitation–exploration trade-off where cognitive processes involved in searching within or between clusters are distinct. In addition, these search processes may also relate to the dual-process model of creativity. The switching component may partly align with the flexibility pathway of the dual-process model of creativity that allows controlled access to broad exploration[113], whereas the clustering component may align with the effortful persistence pathway that provides inhibitory focus necessary for systematic thinking.

The cognitive mechanisms captured by our clustering and switching components can be further discussed in light of the brain's functional connectivity patterns predicting the components. The CPM approach allowed us to identify specific brain connectivity patterns robustly predicting individual differences in each component. These predictions were also generalizable to the individual's resting-state functional connectivity suggesting the robustness of our predictive models. The positive brain model network predicting clustering and switching components involved networks classically involved in creative thinking, namely the ECN, DMN, and salience networks[13,27,87,89], but predictive patterns also clearly differed. The clustering predictive model network predominantly relied on inter-network connectivity between the ECN, DAN, salience, somatomotor and visual networks. The involvement of the salience and ECN is consistent with patient studies showing that these networks are critical for category-fluency tasks[83,84]. Here, the ability to retrieve multiple semantically related concepts seems to be supported by interactions between control and attentional networks with visual and motor regions. Importantly, the temporal pole, a region considered as a hub region of semantic knowledge[30,114] participated in the positive model network through its connection with the salience network. The involvement of task-positive networks and their interaction with a semantic memory region would be consistent with the proposed role of goal-directed attention in the clustering component allowing a sustained processing and a focused search in semantic memory[115].

Key nodes of the positive clustering model network were related to the ECN. Although consistent with our interpretation of the clustering component above, this result was not in line with Troyer et al.[46,51] who related clustering to temporal regions of semantic memory and switching to frontal control processes during a category-fluency task. It is possible that our findings differ from Troyer et al.'s hypothesis[46,51] because a fluency task starting with an ambiguous word, even with an unconstrained instruction, requires more control to stay efficient and focused[116], and indeed, processes that are captured by clustering and switching measures vary depending on fluency task types[46,54]. However, the alignment of clustering and switching in fluency tasks on temporal and frontal functions has already been contested. For instance, in ref. [46], patients with temporal lobe lesions were impaired on category-fluency switching in comparison to controls. Reverberi et al.[85] showed that the performance of lateral frontal patients had a disorganized search strategy but no switching deficit. In addition, a brain stimulation study targeting the dorsolateral prefrontal cortex showed an impact on clustering (and not switching) during a category-fluency task[117]. These findings suggest that the processes reflected in the clustering measure are not purely associative and may involve additional control processes that focus attention. Overall, it is likely that our

clustering component captures not only semantic memory associations but also top-down regulation of semantic search that may inhibit more divergent exploration.

In contrast to the positive clustering model network, the predictive patterns of the switching mainly involve DMN-related brain regions, in addition to regions of the salience and visual networks. The most densely connected brain regions in the switching model network included several regions of the DMN, in particular in the inferior parietal lobule bilaterally, retrosplenial cortex, and medial prefrontal cortex. The involvement of the DMN in our task was expected since previous studies have shown that the generation of spontaneous associations involves brain regions of the DMN[15,18,19,118]. DMN-related brain regions are thought to be essential for the spontaneous processes[119] contributing to creative abilities, especially the medial prefrontal cortex[89,118]. In addition, the DMN has been associated with several aspects of cognition involving heteromodal memory retrieval[120,121] and semantic goal maintenance[122]; its interaction with the ECN correlates with a task similar to the CAT[123]. Importantly, the connectivity predicting switching involves connections of DMN regions with other networks, in particular with ECN regions and the temporal pole, rather than within-DMN connections, suggesting that switching requires interactions between the DMN and other networks. The connectivity between DMN and ECN uniquely predicted switching, not clustering in PolyFT. This finding is consistent with the behavioral findings suggesting that interactions between associative and control processes support flexible search in memory. This interpretation is also in line with previous neuroimaging findings showing that DMN-ECN interactions reflect the top-down regulation of spontaneous and self-generated forms of cognition, with ECN guiding and constraining the DMN-related generative processes[15,89,124,125]. Their link with the salience network may allow them to trigger the switching between the engagement of controlled and associative modes of thought, supported by ECN and DMN, respectively[13,89,125,126].

Also aligned with this hypothesis, the left inferior frontal gyrus, a critical region of the semantic control system[29,30,37,60,86,93], plays a role in the positive switching predictive network, through its connectivity with the DMN, salience and visual networks, and with the temporal regions of the ECN. This result is further in line with a series of studies demonstrating the critical role of the left inferior frontal gyrus in fluency tasks[84,127], in controlled semantic retrieval and selection[59,61,128,129], and in resolving competition between incompatible representations[127]. How the ECN regions based on the Schaeffer parcellation[103] align with the semantic control system remains to be directly addressed (see for instance refs. [31,130]). In addition, echoing the correlation between the switching component and CAT, the predictive network includes node regions of the ECN and DMN that have been shown critical for CAT performance in patients[15] in the rostral inferior frontal gyrus and medial prefrontal cortex.

The involvement of the ECN in the prediction of switching performance is also consistent with the correlation of the switching component with executive tests that typically implicate regions of the ECN[84,131–133]. Overall, these results are in line with several studies showing that since ambiguous words activate multiple meanings, the processing of these words involves processing demands related to the executive semantic system[93,94,134–136].

In addition to regions of the DMN and ECN, a posterior region of the left orbital frontal cortex (limbic network) is a high degree node in the positive switching predictive model. This brain region has been associated to inhibitory control[84,137], and damage to this region had been associated with difficulties in suppressing inappropriate behaviors and perseverative responses[138,139], functions that likely play a role in avoiding inappropriate or perseverative responses during PolyFT performance.

The negative model networks predicting both clustering and switching do not involve attentional or cognitive control regions such as the DAN or the lateral frontal cortex. On the contrary, higher connectivity in ECN and DAN related to higher clustering and higher AUT performance. Hence, our results do not support the hypothesis of a release of regulatory control leading to higher creative abilities[140] (see also ref. [141]).

Overall, switching in PolyFT may be supported by interactions between DMN and ECN or semantic control system (as well as with their interaction with the salience, visual and somatomotor networks), highlighting the contribution of controlled processes to explore diverse semantic fields and the importance of the DMN for self-paced switching behavior. Interestingly, switching related to salience-DMN connectivity whereas clustering was related to salience-ECN connectivity, which may help to further tease apart the involvement of Salience-DMN-ECN connectivity patterns observed in more complex forms of creative cognition (see refs. [87,89]). Importantly, we find evidence of ECN activity in PolyFT for both clustering and switching. For clustering, the ECN-salience connectivity and activation of the dorsal attention network may indicate a focused attention. For switching, ECN-DMN connectivity (as well as with their interaction with the salience, visual and somatomotor networks), may support controlled processes to explore diverse semantic fields. Hence, behavioral and brain correlations converge in suggesting that the PolyFT task effectively captures the dual processes proposed for semantic memory search[64], and that both processes involve controlled processing but in different ways. These findings lend further support to semantic memory search being a two stages process (see refs. [47,65,66]).

Some limitations of this study should be mentioned. First, the sample is relatively small, especially for conducting the CPM approach. However, the results are robust to statistical procedures based on permutation testing and cross-validation. The internal validation using the resting-state data of the same participants may be influenced by the similarity between task and rest-based functional connectivity data[142]. Third, some correlations did not survive multiple comparisons likely due to a lack of power. Future studies should consider a higher number of participants and external validations to further support the robustness of our results. In addition, replication of these results with more cue words for the PolyFT task would be useful to draw more general conclusions on the underlying processes. Fourth, it is possible that the RJT involves executive processes recruited for judging the relatedness between word pairs, which may have influenced the CPM models. However, SemNet metrics based on the RJT ratings did not correlate with the executive tests we used, and they have been previously validated in several studies as assessing the structure of semantic memory[24,26,27,112]. Furthermore, Benedek et al.[24] have found that SemNet structure and executive processes independently predicted individual differences in creative thinking. Fifth, when exploring the UUN graph metrics, the threshold applied to the networks could yield binary graphs with different densities, which may impact the graph metrics. Future work should address this issue by developing novel approaches to explore SemNet metrics independent of the network density. Finally, it is important to note that our results reflect brain-behavior relationships that do not address whether the individual differences in brain network architecture are the cause or the effect of individual differences in semantic search components related to creative ability.

In conclusion, the current findings contribute to better understand the cognitive processes of memory search that relate to creativity and their neural correlates. Clustering and switching

measures were related to divergent and convergent creativity tasks, respectively, and were predicted by distinct brain connectivity patterns. The switching component correlated with semantic memory structure and executive functions, and was predicted by brain functional connectivity between and within the DMN, ECN, and salience networks. This suggests that flexibility in memory search relies on interactions between spontaneous and controlled processes guiding the search. Contrary to our initial expectations, the clustering component did not correlate with semantic memory structure and its predictive connectivity patterns relied on interactions between ECN, salience, and attentional regions. Thus, clustering was not limited to associative memory processes, but also likely captured attentional focus allowing persistent memory search behavior, and thus may reflect a distinct, controlled semantic search mechanism that complements switching in search of remote, yet relevant associations. Together these findings shed light on the neurocognitive mechanisms allowing efficient and flexible semantic memory search in support of creative cognition, based on alternations between focused attention and exploratory search.

## Methods

**Participants**. Behavioral and MRI data from 93 healthy participants (43 women, mean age = 25.6 years, SD = 3.83) were collected in total. Data from seven participants were excluded due to a brain abnormality revealed by the MRI acquisition (n = 6), and for falling asleep during the experiment (n = 1). The final sample consisted of 86 participants (43 women, mean age = 25.5 years, SD = 3.48), who took part in a larger MRI study on creative thinking[27]. All participants were French native speakers, right-handed, with normal or corrected-to-normal vision and with no neurological disorder, cognitive disability or medication affecting the central nervous system. All gave informed written consent and received monetary compensation for their participation. The study was approved by the approved French ethics committee "CPP Sud Mediterranee IV" and we have complied with all relevant ethical regulations.

**General procedure**. After being informed on the experiment and overall visit, the participants first completed the relatedness judgment task (RJT) inside the MRI scanner. Then, outside the scanner, they completed two creativity tasks (AUT and CAT), neuropsychological tests assessing executive processes relevant to creativity, and the PolyFT task. The tasks are detailed below.

**Ambiguous word-fluency task (PolyFT)**. In the PolyFT task, participants are required to name all the words that they could think of as associated with a given cue word (i.e., single-word associations) within one minute. As cue words, we selected three ambiguous, polysemous French words: somme (sum), glace (ice), and rayon (ray; note that English translation does not convey all the meanings of these words in French), that have high lexical frequency (> 20 occurrences per million in a large corpus according to the French lexicon project, http://www.lexique.org/) and at least five different meanings according to the French dictionary (Supplementary Table 1). The list of the different meanings of each cue was determined using a French linguistic resource for research (Centre National de Ressources Textuelles et Lexicales; https://www.cnrtl.fr/). For each cue word, participants generated words associated with the cue word, without being informed that the words had different meanings. The cue words were read out by the experimenter and also presented visually on a paper sheet. The participants gave their responses orally, which were written down by the experimenter. Participants' responses were cleaned (orthographic errors and typos) and coded by EV as referring to one of these meanings (Supplementary Table 1).

For each participant, and for each cue word, several measures were computed: (1) fluency (the total number of different words generated during the time allowed); (2) the number of different meanings the responses refer to; (3) the number of switches between meanings quantified as the number of time two successive words refer to two different meanings; (4) the biggest cluster size, that was the largest number of successive responses that refer to the same meaning during the task; and (5) the rank of the first switch, that was the rank of the first response when the participant referred to a new meaning. Because there is a debate on how to best measure switching[51,53,54] we measured both the total number of different meanings and every transition from one meaning to another and used a principal component analysis approach (see below). We also measured the rank of the first switch in an attempt to capture how much participants stayed in the same meaning before considering alternative ones. The measured variables were calculated on each cue word separately and averaged across the three cue words for each participant.

**Principal component analysis of the ambiguous word-fluency task measures**. We ran the PCA on the five PolyFT measures averaged with oblique rotation (direct oblimin) to rotate loadings and identify the variables contributing the most to each component in the 86 participants. The PCA analysis was based on a correlation matrix. Components with eigenvalues over Kaiser's criterion of 1 were kept with confirmation of their relevance using the scree plot. Sampling adequacy for the analysis was assessed with the Kaiser–Meyer–Olkin measure (standard threshold = 0.507).

**Creativity assessment with the alternative uses task (AUT)**. In the AUT, participants were given three minutes to continuously generate alternative uses for each of three common objects (i.e., a tire, a bottle, and a knife). The name of the object was pronounced by the examiner and displayed on the screen for 3 min. The participants wrote down their responses on the computer using the keyboard and could see all their responses during the task period. At the end of the three minutes, participants were asked to select their two most creative ideas for each object (top-2)[105,106]. All the generated responses were coded in order to homogenize similar ideas that were differently formulated. Then, classical scores for divergent thinking tasks were computed. The fluency quantified the total number of responses generated by the participants summed across the three objects (AUT-fluency). The uniqueness score (AUT-uniqueness) was the sum of the number of ideas across the three objects that were generated by less than 5% of the other participants. The overall frequency of responses (AUT-commonness) quantified the frequency with which each idea was generated within the group averaged across all responses and objects for each participant. Finally, five independent judges who were lab members familiar with creativity ratings were given written instructions to rate the creativity of each of the top-2 responses (AUT ratings)[105,106]. They used a Likert scale from 0 (not creative) to 4 (highly creative). The inter-rater reliability showed an intraclass correlation coefficient equal to 0.74. The database to compute the AUT-uniqueness and the AUT-commonness included eight additional subjects who performed the set of creativity tasks but not the PolyFT.

**Creativity assessment with the combination of associates task (CAT)**. The CAT is an adaptation of the Remote Associates Test[9], a classical task assessing convergent thinking in creativity. In the CAT, participants completed 100 trials composed of triplets of cue words that at first glance seem to be unrelated. On each trial, they were asked to provide a solution word that relates to each of the cue words. In developing the CAT, we varied the semantic distance between the cue words and the solution, as described in previous articles[15,96]. In close trials (n = 50), the solution had a semantically close relationship to the cue words, whereas in distant trials, the solution is remotely related to the cue words (n = 50). On each trial, participants were given 30 s to find a solution. They were asked to press the space bar as soon as they thought of the solution, and to write it down on the computer within 5 s using the keyboard. Next, they were asked to report whether the solution came to their mind with a feeling of insight or Eureka, i.e., when the solution arose suddenly and effortlessly to mind[8]. They were given five additional seconds to press the "v" key if they found the solution with an insight, or the "n" key otherwise. We quantified the accuracy in CAT as the total percentage of correct responses (CAT-CR). To differentiate the performance of the participants in the close versus the distant trials, we calculated an index as the difference in accuracy in the close and distance conditions, divided by the overall accuracy (CAT-index). This index reflects the ability to solve the more distant trials when controlling for performance in the closer trials: the lower the CAT-index, the higher the creativity abilities. Finally, CAT-eureka score was the percentage of reported Eureka among correct trials.

**Relatedness judgment task (RJT)**. The RJT task was used to estimate the individual SemNets[24–26]. This task consists in judging the semantic relationship between all possible pairwise combinations of 35 words controlled for linguistic properties and semantic distance between all word pair combinations. We selected the 35 words of the RJT task based on their linguistic properties and the distribution of the theoretical distance (strength of association) between each pair of words (see ref. [25] for details). In total, the participants performed 595 RJT trials (Fig. 2a). For each trial, the word pair was presented on the screen along with a visual scale below going from 0 to 100. Participants were instructed to rate the relationship or semantic association between the two words using a slider. They were encouraged to use all the values of the scale for their judgments going from 0 (totally unrelated words), to 100 (totally related words). Each trial was composed of a 2-s thinking period and a 2-s response period. The thinking period started with the display of the words and the visual scale. Participants were instructed to think about the relatedness between the words. After this two-second thinking period, the slider appeared in the middle of the scale. Participants could then freely move the cursor on the scale using a mouse and validating their response by a left-click. This response period ended with the validation or when the 2 s of the response period had elapsed. The final position of the slider in the scale after validation was considered as the semantic relatedness rating. When participants did not validate the trials (3.5% of the trials across participants), the final position of the slider at the end of the response time was considered as the semantic relatedness rating. Note that in all trials (including unvalidated ones) the slider had been moved by

the participants from its initial position. An inter-trial interval jittered from 0.3 to 0.7 seconds (mean 0.5; interval: 0.05) separated two successive trials. Trials were grouped in 24 blocks of 25 trials each (except the last block that lasted 20 trials) and organized into six runs of four blocks each. Blocks were separated by 20 seconds rest periods, and runs by self-paced rest periods. Each run lasted around nine minutes. Trials were pseudorandomly attributed to blocks with the constraint that the distribution of the theoretical distances between word pairs was equivalent between blocks (see refs. [25,27] for details).

**Estimating individual semantic networks based on the RJT task.** We used the ratings given by the participants in the RJT task to build their individual SemNets. For each participant, we built a 35 words × 35 words adjacency matrix which cells corresponded to the rating between the node pairs. From the adjacency matrix, we built the WUN and the UUN SemNets. The WUN was composed of 35 nodes (i.e., RJT words), and the edges connecting the nodes were weighted by the participants' ratings. The UUN was composed of the same 35 nodes, and the edges were thresholded so that only ratings higher than 50 (middle of the scale) were kept and then binarized. For the UUN, we selected the threshold value of 50 (at the middle of the visual scale, hence retaining associations between concepts only when rated as moderately to highly related) based on previous studies[24,25] that showed how UUN metrics thresholded in this way predicted creativity. In this way, only associations between concepts were kept when they had been rated as moderately to highly related. The selection of this threshold was validated by an additional analysis in which we computed each SemNet metric for a range of thresholds and used the area under curve for subsequent correlations with behavior[143] (see Supplementary Note 1).

**Computation of the individual semantic network metrics.** We built the UUN and the WUN graphs separately and computed the network metrics[97–100]. Eff is calculated as the inverse value of the average of shortest path lengths. CC is computed for each node as the proportion of neighbors that are neighbors to each other. Within the SemNets, higher clustering represents more connected or related concepts. Q is measured as the level of division of the network into smaller sub-networks or communities. In SemNets, these communities can represent different semantic categories. The computation of the SemNet metrics were performed in Matlab, via the Brain Connectivity Toolbox[100].

**Executive control assessment with the digit span test.** The digit span test of the Wechsler Adult Intelligence Scale (WAIS)[144] was used to assess working memory ability. We used the criteria of performance of this task based on the WAIS manual. This task consisted of two parts in which participants repeated a string of numbers that increased in size. In the first part (forward-span), participants were instructed to repeat 16 different strings of numbers going from 2 to 9 digits in the same order. In the second part (backward-span), they were instructed to repeat 14 different strings from 2 to 8 digits, but reversely. Participants were given with a pair of strings as examples before the actual tasks. To quantify the total score, we gave 1 point for each correct recall. Both parts were quantified separately.

**Executive control assessment with the trail-making test.** The trail-making test (TMT) assesses set-shifting[145]. This task is composed of two parts. In the first part (A), participants were presented with numbers distributed randomly on a sheet of paper. They were instructed to link the numbers (from 1 to 25) in increasing order with a pen, as fast as possible. In the second part (B), participants were presented with numbers (from 1 to 13) and letters (from A to L) distributed randomly within the sheet. They were instructed to link the numbers and letters alternately in increasing order, as fast as possible. The time to complete each part was measured. We quantified the difference between the time to complete the second minus the first part (TMT-shifting).

**Executive control assessment with the category and letter-fluency tasks.** Participants performed two fluency tasks measuring broad retrieval abilities, a category-fluency task (with the category animals)[146], and a letter-fluency task (with the letter 'F'). For each task, participants were given two minutes to continuously generate as many words as possible. We recorded and transcribed the responses of each participant. Based on ref. [146], we quantified the total number of responses generated by the participants for each task separately.

**Executive control assessment with the Stroop test.** The Stroop test[147], measuring inhibition, was based on the French version in ref. [148]. In the first part, participants were presented with 100 written names of different colors and were instructed to read these names aloud. In the second part, participants were presented with 100 colored squares and were instructed to name the color of the ink. In the third part, participants were presented with 100 written names of colors with different color ink. Participants were instructed to say the color of the ink aloud. Each part was timed and the total time to complete each part was recorded. Based on ref. [148], we quantified the interference effect (Stroop-interference) as the difference in time to complete the third (ink naming with interference) and the second part (color naming).

**Statistics and reproducibility.** We explored the Spearman correlations between all behavioral measures and the PolyFT components of all participants ($n = 86$). We ensured that the number of the unvalidated trials during the RJT task was not correlated to any of the PolyFT components, and the measures of creative abilities, SemNet metrics, and executive function abilities. We used an FDR approach to correct for multiple comparisons. We reported the $p$-values for all significant correlations both before and after the correction for all the comparisons that were performed, as a compromise between overly conservative and liberal approaches.

**MRI data acquisition and preprocessing.** Whole-brain imaging was acquired on a 3T MRI scanner (Siemens Prisma, Germany) with a 64-channel head coil. The fMRI data was acquired during the RJT runs using multi-echo echo-planar imaging (EPI) sequences. Each run consisted of 335 volumes acquired with repetition time (TR) = 1600 ms, echo times (TE) for echo 1 = 15.2 ms, echo 2 = 37.17 ms and echo 3 = 59.14 ms, flip angle = 73°, 54 slices, slice thickness = 2.50 mm, isotropic voxel size 2.5 mm, Ipat acceleration factor = 2, multi-band = 3 and interleaved slice ordering. In addition to functional imaging, a T1-weighted structural image was acquired using TR = 2300 ms, TE = 2.76 ms, flip angle = 9°, 192 sagittal slices with a 1 mm thickness, isotropic voxel size 1 mm, Ipat acceleration factor = 2 and interleaved slice order. In the last part of the session, we acquired resting-state fMRI data for 15 min. No volume was discarded from any of the fMRI data since the recording did not contain dummy scans.

Functional volumes of each run and for resting-state data were first despiked, slice timing corrected and realigned to the first volume (computed on the first echo) using the afni_proc.py pipeline from the Analysis of Functional Neuroimages software (AFNI; https://afni.nimh.nih.gov)[149]. In a second step, the data were denoised using the TE-dependent analysis of multi-echo fMRI data (TEDANA; https://tedana.readthedocs.io/en/stable/), version 0.0.9[150,151]. The TEDANA pipeline consisted of an optimal combination of the echo time series followed by the reduction of the data using PCA and independent component analysis (ICA) to decompose the multi-echo BOLD data, and classify the BOLD components as BOLD or non-BOLD. The removal of the latter eliminates the thermal and physiological noise including the artifacts generated by the movements, respiration, and cardiac activity. The advantage of acquiring multi-echo EPI sequences is the possibility of assessing the BOLD and non-BOLD signal through the ICA-based denoising method, improving the reliability of the functional connectivity-based measurement[152]. In the last step of the preprocessing, the data were co-registered on the T1-weighted structural image using the Statistical Parametric Mapping (SPM) 12 package running in Matlab (Matlab R2017b, The MathWorks, Inc., USA) and normalized to the Montreal Neurological Institute (MNI) template. To spatially normalize the fMRI data, we used the transformation matrix computed from the normalization of the T1-weighted structural image, using with the default settings of the computational anatomy toolbox (CAT 12; http://dbm.neuro.uni-jena.de/cat/)[153] implemented in SPM 12.

No participant was removed due to excessive head motion within a single run, defined a priori as >2 mm translation or >3° rotation, and no participant had mean FD > 0.5 mm. To covary out the task-related signal from each run, the denoised and normalized fMRI data were entered in a general linear model in SPM. We regressed out of the BOLD signal 24 motion parameters (standard motion parameters, first temporal derivatives, standard motion parameters squared, and first temporal derivatives squared) and the onsets and durations of each task-related events (reflection period, response period, inter-trial interval, cross-fixation periods, and change of the cross-fixation color). We then standardized and detrended the residuals of the GLM for each run, and concatenated the six runs, removing the between runs rest periods. These preprocessed data were used in the subsequent analyses.

**Connectome-based predictive modeling (CPM) of PolyFT components.** We used the CPM approach to analyze how cognitive components of the PolyFT task rely on whole-brain functional connectivity. We used CPM with a cross-validation approach since it is a more conservative way to infer a brain-behavior relationship than the typical correlation approach, allowing to increase the likelihood of replication in future studies and preventing the overfitting of the data[102]. The functional connectivity matrices for the task and resting-state data of each participant were computed using Nilearn v0.3[154] in Python 2.7[155]. We defined regions of interest (ROIs) based on the Schaefer brain atlas[103] which includes 200 ROIs of 2-mm dimensions distributed into 17 functional subnetworks distributed on eight main functional networks. We extracted and averaged the BOLD signal for each ROI and performed Pearson pairwise correlations. As a result, we obtained a 200 × 200 matrix for each participant that were Z-Fisher transformed and rescaled by the maximal weight to be in the range of −1 to 1 for the subsequent analyses. The CPM analysis[102] consisted of five steps, and a validation process. Since we used a leave-one-out validation, the first three steps were performed in N-1 participants, and the fourth step was performed in the left-out individual.

In the first step, we selected the behaviorally relevant edges of the brain connectivity individual networks. We selected the connections in the functional connectivity matrix ($z$-scored connectivity values between ROIs) that significantly correlated with each PolyFT component (threshold $P < 0.01$) either positively (the positive model network) or negatively (the negative model network) across participants. We used Spearman correlations to avoid the possible influence of

outliers in the predictions. The ROI pairs (i.e., brain links) correlated with each component formed the positive and the negative model network for the fluency and flexibility components. In the second step, we estimated the connectivity strength of the positive and the negative model brain network, separately. In the third step, we built a linear model using the individual connectivity strength in the positive and negative model networks as predictors and the PolyFT component as the outcome. To further ensure that the results are not biased by movement during fMRI, the mean FD (sum of the absolute values of the derivatives of the six realignment parameters) was included in the predictive model. We also tested whether the movements during fMRI impact the results by running the same analysis without including mean FD in the model (see Supplementary Note 3). In addition, the mean FD values had no correlated with any of the PolyFT components (clustering: $r_s = 0.016$, $P = 0.89$; switching: $r_s = -0.051$ $P = 0.64$). The fourth step was a leave-one-out validation. We applied iteratively the predictive linear model built on N-1 participants to the left-out participant. This allowed us to obtain a predictive value of the PolyFT component for each participant. The final step tested the prediction of the linear model by running Spearman correlations between the predicted and the observed values of the PolyFT component. Hence, we ran a separate analysis for the PolyFT components. Since we used within-data set cross-validation, when Spearman correlations were significant, it was necessary to evaluate the predictive power of the predictions using permutation testing. To this end, we randomly shuffled the observed values 1000 times, and we ran the pipeline of our predictive model using the new random data. Thus, we generated an empirical null distribution and estimated the distribution of the test statistic given by the correlation between predicted and observed values. The CPM analyses were performed using Matlab Statistical Toolbox (Matlab R2017b, The MathWorks, Inc., USA).

We explored the functional anatomy of the positive and negative model networks of each score by localizing and describing the main nodes and links of the significant model networks. Since we employed a leave-one-out method, for each iteration the number of links can be slightly different. For a better interpretation of the results, we considered the links that were significant in all iterations. We examined the distribution of the connections at the region level (between and within brain lobes) and at the intrinsic network level (within and between the eight main functional networks defined by the Schaefer atlas). Finally, we identified the highest degree nodes (i.e., ROIs), which are the nodes with the highest number of connections within the predictive networks. With the leave-one-out approach, the model networks were estimated $N$ times on $N - 1$ participants, and thus each iteration likely resulted in slightly different model networks. To obtain a reliable representation of the predictive model networks, we selected for this description the connections in the model networks that were shared between all iterations. The data visualization and plots were performed using BioImage Suite Web 1.0 (http://bisweb.yale.edu/connviewer), BrainNet viewer[156] (http://www.nitrc.org/projects/bnv/) in Matlab, and custom scripts in RStudio version 1.3.1056.

**Internal validation: prediction of PolyFT components from resting-state functional connectivity**. As an internal validation test, the predictive models built based on the task-based functional connectivity were applied to the participant's resting-state data (i.e., the strength of functional connectivity within each participant's positive and negative model networks during resting-state acquisition). Similar as for the task-based analyses, the mean FD was included in the model, and the Spearman correlations between the predicted and the observed PolyFT components were computed to evaluate the prediction.

**Reporting summary**. Further information on research design is available in the Nature Research Reporting Summary linked to this article.

## Data availability
All data needed to evaluate the conclusions in the paper are present in the paper and/or the Supplementary Information, or are available at https://osf.io/uktjm/?view_only=3ab90072a7804e08ad80d2b8c45ced19.

## Code availability
Analyses were conducted using open software and toolboxes available online as described in "Methods" (SPM 12: https://www.fil.ion.ucl.ac.uk/spm/software/spm12/; AFNI: https://afni.nimh.nih.gov); Nilearn: https://nilearn.github.io/stable/index.html; TEDANA: https://tedana.readthedocs.io/en/stable/; CPM: https://www.nitrc.org/projects/bioimagesuite/; Network metrics computation: https://sites.google.com/site/bctnet/Home/functions). Any custom scripts written for this paper are available upon request from the corresponding author.

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

## Acknowledgements

The research was supported by "Agence Nationale de la Recherche" [grant numbers ANR-19-CE37-0001-01] (E.V., M.B., and Y.N.K.), the "Fondation pour la recherche medicale" [grant number: DEQ20150331725] (E.V.), and received infrastructure funding from the French program "Investissements d'avenir" ANR-10-IAIHU-06 (E.V.). This work was also funded by Becas-Chile of ANID-CONICYT (MOT). The funder had no role in study design, data collection and analysis, decision to publish, or preparation of the manuscript. This work was carried out in the PRISME and CENIR facilities of ICM. We gratefully acknowledge Karim N'diaye, Benoit Beranger, and Romain Valabregue for their help in the data collection. We thank the participants for making this work possible.

## Author contributions
E.V.: conceptualization, methodology, writing—original draft, writing—review and editing, supervision, project administration, resources, and funding acquisition. Y.N.K.: conceptualization, methodology, writing—review and editing, and funding acquisition. M.B.E.N.: conceptualization, methodology, writing—review and editing, and funding acquisition. M.O.T.: methodology, software, formal analysis, investigation, writing—original draft, writing—review and editing, visualization, validation, and data curation. M.B.E.R.: software, investigation, and writing—review and editing. T.B.: formal analysis and writing—review and editing. T.H.: conceptualization, writing—review and editing. S.B.: investigation, writing—review and editing. J.B.: Investigation and writing—review and editing.

## Competing interests
The authors declare no competing interests.
