## [Peer Review File · Communications Biology]

Reviewers' comments:

Reviewer #1 (Remarks to the Author):

This is an interesting manuscript that examines the cognitive and neural correlates of semantic memory search related to creative ability. The study presents a novel task designed to distinguish between two components related to semantic search: clustering and switching. Findings shed light on neurocognitive correlates associated with semantic memory search and provide helpful tests of existing theory with implications for future work in this area.

There are a number of strengths to this manuscript:

- The procedure contains a substantial number of tasks which provides a comprehensive assessment of neurocognitive correlates of clustering and switching
- Materials to evaluate the conclusions of the manuscript are made available online

There are a number of places where the manuscript may be improved:

- Modularity, Q , is used as a measure of the level of segregation within semantic networks. Given that the Brain Connectivity Toolbox was used, I am assuming that a Louvain-like locally greedy algorithm was used but that would be good to specify. The modularity heuristic is non-deterministic and, in typical applications, the algorithm is iterated multiple times (e.g., 100 times) and the average Q from these iterations is taken. It is unclear how many iterations were undertaken in the current application.
- Relatedly, Q is strongly dependent on network density/strength. As such, it would be important to control for network density when examining associations between outcomes of interest and Q .
- Participants completed a relatedness judgement task while undergoing fMRI. Although the fMRI data analyzed represents task data, the task is regressed from the BOLD data. It would be helpful to include a rationale for why this residualized task data was used to compute functional connectivity rather than resting state data (which was collected but not reported on).
- Given the limitations of the sample size with reference to the CPM approach noted in the discussion section, it would be helpful to provide a rationale for why the CPM approach was taken instead of more typical approaches wherein whole brain correlations are computed (e.g., correlation approaches often overfit the data and limit generalizability to novel data; Shen et al., 2017 Nature Protocols)
- In the RJT task, participants had 2 seconds to report their answer. Did this lead to missing data or was no response taken to be a response at the middle of the scale (i.e., the default starting location)? If missing data was an option, how much missing data was there and how was it treated?

Minor points

- Unweighted undirected networks were thresholded so that only ratings higher than 50 (middle of the scale) were kept and then binarized. Providing a rationale for this threshold would be helpful as would robustness analyses to examine the extent to which associations change depending on the level of the threshold. However, a strength of the current approach is the use of weighted undirected networks that do not require arbitrary thresholding. As such, I don't think additional sensitivity analyses are necessary, a sentence about why a threshold at the middle of the scale is reasonable would suffice.
- There are moments where the framing of the purpose of the study does not map appropriately onto the methods. For example, "Assessing functional brain connectivity associated with individual differences in clustering and switching may provide a useful approach to capture the brain systems that underlie distinct memory search components". I think the language here (and elsewhere) could be tempered to indicate that neural correlates of between-person differences in distinct memory search components are being revealed by the methods.

Reviewer #2 (Remarks to the Author):

Ovando-Tellez and colleagues employed a series of behavioral assessments and a novel fMRI paradigm - relatedness judgment task - to tease out latent processes during creative semantic

search. The work is highly innovative and comprehensive, and I thoroughly enjoyed reading the manuscript. However, there are a few concerns that weaken the significance of the study, as detailed below.

Major:

1. RJT task. Each RJT trial was composed of a two-second thinking period and a two-second response period, after which there was a 0.2-0.7s fixation period. The participants could freely move the cursor on a slider and validate their response with a left click. What if participants failed to respond within the given window? If it did occur, how did the subsequent analysis handle missing trials? Did the task performance drive the prediction? Readers could also benefit from a diagram of the RJT in Fig. 1 since this is the main fMRI task. Besides, have authors looked at the main activation effect of RJT?
2. CPM analysis. In the section "Connectome-based predictive modeling (CPM) of AmbiFlu components," authors included mean FD in the predictive model to remove the influence of head motion (lines 478 – 480). This is confusing since including mean FD in the predictive model potentially exacerbates the problem if mean FD correlated with AmbiFlu components. If authors want to rule out the head motion, authors can run CPM without mean FD and later conduct a partial correlation analysis between predicted and true AmbiFlu scores controlling for motion. Please also report the negative CPM network. The author wrote, "Note that the negative model networks only included a small number of links." For the clustering component, the positive network contains 43 edges while the negative network has 14, and I would not call it a small number. The negative network could be used to examine the hypofrontality hypothesis that a loosened top-down regulatory control boosts creativity abilities (Chrysikou et al., 2013). It would be interesting to see if the negative switching model consists of negative links in DAN (or a decoupled left IFG) to enable less constraint exploration (Chrysikou et al., 2013; Kleinmintz et al., 2018; Xie et al., 2021)?
3. Graph analysis. When calculating the semantic network metrics for binary graphs, authors chose an arbitrary threshold of 50 to binarize the weighted semantic graph. This is potentially problematic since using the same threshold across subjects would yield binary graphs with different densities, altering graph metrics. For example, the authors claimed, "These results suggest that individuals with higher switching during the AmbiFlu task had a more efficient SemNet, that was more connected and less modular" (Line 548-550). Could a more connected and less modular network arise simply because the participants rated everything as highly related, leading to a denser graph? Alternatively, instead of using a fixed threshold across subjects, authors could keep 50% of the edges of the semantic network, which would make the graph metrics more comparable across subjects. Also, it would be interesting to run a consensus clustering using consensus_und.m in brain connectivity toolbox on the individual-level communities to obtain a group-level community, just to see if the identified communities correspond to any meaningful categories.
4. Multiple comparisons. Line 522-524: "The correlations between clustering and both AUT-fluency and AUT-uniqueness remained significant after FDR correction for multiple comparisons." How many comparisons were corrected for? All the pairwise correlations in Fig. 2? Given how many correlations were computed in Fig. 2 ($20 \times 19 / 2 = 190$), it seems unlikely that all p s < 0.01 passed FDR. Also, it may be clearer if authors just report significant p -values after correction, instead of first reporting raw p -values and then stating what remained significant after correction.
6. Figures 4 and 5 could be improved. The table in Fig 4(A) and Fig. 5(A) should have "lobe" instead of "network" on top. I don't see any additional benefits of including panel (A) since all discussion was based on the functional networks. Besides, how was the percentage calculated, and why do they not sum up to 100? Panel B in Fig 4 and 5 is too densely connected and is not very informative. Perhaps show only a few highest-degree nodes. Show the full node-to-node CPM model instead of summed over the network may also help. The degree itself did not mean much without knowing how many nodes each network has.
7. The authors extracted two latent components from the AmbiFlu task, i.e., clustering and switching, and introduced clustering as convergent and deliberate exploitation (more similar to CAT) and switching as a divergent and spontaneous exploration (more similar to AUT). However, the behavioral results did not quite match that framing, as neither switching correlated with AUT nor clustering correlated with CAT performance, which made me wonder about the validity of these latent components. One potential improvement could be the semantic distance across generated words

(Beaty and Johnson, 2021; Olson et al., 2021), as some clusters/meanings must be harder to reach than others. Another potential improvement is that instead of using unsupervised data-driven PCA followed by CPM, authors can use methods such as partial least square regression to jointly investigate the behavioral and neural signatures of creativity and then identify the latent behavioral components, for example (Kebets et al., 2019).

8. The organization of the materials and methods section is not easy to follow. The authors conducted multiple behavioral assessments and creative tasks and gave detailed descriptions to all of them in the main text, including many widely-used tasks such as Stroop, AUT, and CAT among others. However, it appeared that the main tasks used in the main analysis were relatedness judgment task (RJT) and ambiguous word fluency (AmbiFlu) task. Thus, the authors could consider only keeping the detailed description of RJT and AmbiFlu in the main text, and briefly introducing other behavioral assessments and moving details to the supplementary materials. Moreover, the "Principal component analysis of the ambiguous word fluency task measures" should be in the Results section. Also, with so many acronyms, the authors should add a glossary of terms, and group them by categories, such as executive functions, network metrics, and creativity.

Minor:

1. Preprocessing does not contain either temporal filtering or detrending. In theory, low-frequency drift or high-frequency cardiac, respiration, and motion-related noise should be rejected in the multi-echo denoising, but considering the data-driven nature of the denoising strategy, I wonder if it is better to remove trends and filter out high-frequency signals before GLM analysis.
2. Line 458 should be "pairwise correlations."
3. In the CPM section, the authors stated, "These matrices were Z-Fisher transformed 460 and rescaled by the maximal weight to be in the range of -1 to 1 for the subsequent analyses" (line 460-461). I wonder how the weights are scaled within each connection or participant? Usually, they are z-scored for each connection.
4. Please report the mean FD across subjects, such as mean FD across subjects (how many subjects with mean FD > 0.5mm) and how many frames were potentially contaminated by head motion (i.e., how many TRs had FD > 1mm). I raised this question because no one was excluded due to excessive head motion, which is not very common.
5. Add corresponding names for each component in Table 1.
6. Were there any wrong answers in the AmbiFlu task, i.e., answers deemed unrelated to the cue?

Reference

- Beaty, R.E., Johnson, D.R., 2021. Automating creativity assessment with SemDis: An open platform for computing semantic distance. *Behav. Res. Methods* 53, 757–780. doi:10.3758/s13428-020-01453-w
- Chrysikou, E.G., Hamilton, R.H., Coslett, H.B., Datta, A., Bikson, M., Thompson-Schill, S.L., 2013. Noninvasive transcranial direct current stimulation over the left prefrontal cortex facilitates cognitive flexibility in tool use. *Cogn. Neurosci.* 4, 81–89. doi:10.1038/jid.2014.371
- Kebets, V., Holmes, A.J., Orban, C., Tang, S., Li, J., Sun, N., Kong, R., Poldrack, R.A., Yeo, B.T.T., 2019. Somatosensory-Motor Dysconnectivity Spans Multiple Transdiagnostic Dimensions of Psychopathology. *Biol. Psychiatry* 86, 779–791. doi:10.1016/j.biopsych.2019.06.013
- Kleinmuntz, O.M., Abecasis, D., Tauber, A., Geva, A., Chistyakov, A. V., Kreinin, I., Klein, E., Shamay-Tsoory, S.G., 2018. Participation of the left inferior frontal gyrus in human originality. *Brain Struct. Funct.* 223, 329–341. doi:10.1007/s00429-017-1500-5
- Olson, J.A., Nahas, J., Chmoulevitch, D., Cropper, S.J., Webb, M.E., 2021. Naming unrelated words predicts creativity. *Proc. Natl. Acad. Sci. U. S. A.* 118, 1–6. doi:10.1073/pnas.2022340118
- Xie, H., Beaty, R.E., Jahanikia, S., Geniesse, C., Sonalkar, N.S., Saggar, M., 2021. Spontaneous and deliberate modes of creativity: Multitask eigen-connectivity analysis captures latent cognitive modes during creative thinking. *Neuroimage* 243, 118531. doi:10.1016/j.neuroimage.2021.118531

Reviewer #3 (Remarks to the Author):

This is an interesting and fairly well-written article examining the relationships between semantic

memory and creative thinking. The purpose of the research was to examine how switching semantic categories and clustering in the context of a novel ambiguous word association task is related to creativity in terms of cognitive and neural processing. Among the strengths of the study is the substantial sample size for an fMRI experiment, the individual semantic network metrics, and the novelty of the task design, which is ideally suited to address the research question—as well as being very interesting.

I have a few comments on the manuscript, as follows:

1. The paper is addressing multiple constructs and at many levels. For example, there is reference to clustering and switching, the relationship with these constructs and associative vs. controlled processing, and then the person-specific neural correlates of these systems. I found the introduction to be somewhat confusing in presenting a clear hypotheses on the anticipated relationships among these constructs and a clear rationale of how understanding these relationships elucidates our understanding of creative thinking. The current introduction seems to move back-and-forth from the one level of analysis to the other, without sufficient and clear framing of the study's specific predictions.
2. The connections between the constructs mentioned above and the tasks used in the study add to the confusion mentioned under point #1 above. The links between the constructs and the tasks used to measure them needs to be much better established.
3. In addition to the many levels of analysis, we have—critically—the inclusion of neural data, which yet again adds another layer of conceptual complexity to the project. This is not bad, of course, but it does require a clearer and more streamlined argument in the introduction on how all these levels come together and what they tell us about creativity.
4. It would be helpful to know if the authors expect that there is a 'prototypical' structure for semantic memory that is superseding individual differences.
5. What are the study's predictions? These should be stated clearly and justified by the literature explicitly in the manuscript—as well as how support for one hypothesis over another reveals something important about creativity.
6. The discussion of the results in many places implies causality. However, the present data—in my view—are unable alone to support any causal claims regarding switching/clustering, controlled/automatic processes, and memory search and structure. Careful reconsideration of some of the language would be strongly recommended.

Minor comments:

- I would encourage the authors to consider re-naming the ambiguous words task—AmbiFlu may not be an optimal choice (especially in a pandemic...)
- The plural of corpus is corpora (not corpuses)

In the present document, we respond to each comment of the Reviewers in detail, including indications of how we revised the manuscript and excerpts of the revised document and Supplementary Information where appropriate. Please, note that, considering one of the Reviewer's suggestions, we changed the name of the task we developed. Since the name of 'AmbiFlu' was not an optimal choice in these times of pandemic, the task previously named 'AmbiFlu', now became '**PolyFT**' (polysemous fluency task). Furthermore, we conducted new analyses resulting in substantial revisions to the main text or presented in the Supplementary Material. All revisions that have been made are highlighted in the new version of the manuscript to distinguish them clearly.

Reviewer 1:

This is an interesting manuscript that examines the cognitive and neural correlates of semantic memory search related to creative ability. The study presents a novel task designed to distinguish between two components related to semantic search: clustering and switching. Findings shed light on neurocognitive correlates associated with semantic memory search and provide helpful tests of existing theory with implications for future work in this area.

There are a number of strengths to this manuscript:

- *The procedure contains a substantial number of tasks which provides a comprehensive assessment of neurocognitive correlates of clustering and switching*
- *Materials to evaluate the conclusions of the manuscript are made available online*

We thank the Reviewer for the positive general assessment.

There are a number of places where the manuscript may be improved:

R1 Comment #1: *Modularity, Q , is used as a measure of the level of segregation within semantic networks. Given that the Brain Connectivity Toolbox was used, I am assuming that a Louvain-like locally greedy algorithm was used but that would be good to specify. The modularity heuristic is non-deterministic and, in typical applications, the algorithm is iterated multiple times (e.g., 100 times) and the average Q from these iterations is taken. It is unclear how many iterations were undertaken in the current application.*

For community detection, we initially used the method proposed by Newman (2006), which is implemented in the Brain Connectivity Toolbox via the *community_und* function with 1000 iterations. Based on the Reviewer's comment, we additionally computed modularity using the Louvain-like locally greedy algorithm (*community_louvain*) implemented in the Brain Connectivity Toolbox. Both methods gave similar results and specifically there were no changes to statistically significant findings. The correlations between UUN Q and PCA switching ($r_s = .24$, $p = .03$) and category-fluency ($r_s = -.30$, $p = .005$), and between WUN Q and AUT-commonness ($r_s = .22$, $p = .04$) remained significant when exploring Q via the Louvain-like locally greedy algorithm.

R1 Comment #2: Relatedly, Q is strongly dependent on network density/strength. As such, it would be important to control for network density when examining associations between outcomes of interest and Q .

We agree that Q is strongly dependent on network density. However, as the size of our semantic networks is small (i.e., 35 nodes as compared to brain networks with minimum 200 ROIs), it is difficult to control for the number of edges.

A previous study (Benedek et al., 2017) has shown how controlling for the network density removes the variance of the data that is related to creative abilities. This work showed a significant correlation between creativity and SemNet metrics when network filtering was based on a relatedness threshold, but not when it was based on a fixed edge number or on the analysis of weighted networks.

Hence, we acknowledge the confounding effect of the density when exploring modularity. In addition, since CC and Q did not survive multiple testing correction, we modified the result and discussion section as follows:

Results section (p.11): “These results suggest that individuals with higher switching during the PolyFT task had a more efficient SemNet, that was more connected and less modular” has been changed to “The correlation between Eff in the UUN network and the switching component remained significant after FDR correction. These results suggest that individuals with higher switching during the PolyFT task had a more efficient SemNet”

Discussion section (p.24): “Fourth, when exploring the UUN graph metrics, the threshold applied to the networks could yield binary graphs with different densities, which may impact the graph metrics. Future studies should address this issue by developing novel approaches to explore SemNet metrics independent of the network density”.

R1 Comment #3: Participants completed a relatedness judgement task while undergoing fMRI. Although the fMRI data analyzed represents task data, the task is regressed from the BOLD data. It would be helpful to include a rationale for why this residualized task data was used to compute functional connectivity rather than resting state data (which was collected but not reported on).

Using a whole-brain functional connectivity approach based on residualized task data allowed us to assess brain connectivity in a controlled condition (while all participants are performing the same task) but yet independent of task-related brain activation. As another advantage of using task-based fMRI data (task-related connectivity), it allowed us to explore interindividual differences in participants’ functional connectivity while performing semantic judgments between pairs of words, which may emphasize the network relationships during the task engagement compared to resting-state data. Additionally, resting-state data may be more susceptible to arousal confounds (Vanderwal et al., 2015). Importantly, when looking at the

functional connectivity patterns during task conditions, it improves the predictions beyond resting-state functional connectivity (Cole et al., 2021).

We agree that looking at the resting-state data also is a valuable approach. Therefore, we now additionally examined resting-state functional connectivity patterns, which serves as another test and potential internal validation of our predictive model. This analysis enabled us to explore whether the predictive models built from the task-based functional connectivity data generalize to the prediction of the clustering and switching components of the PolyFT task from participants' resting-state functional connectivity. We use the positive and the negative network models from the task-based functional connectivity data from all participants to build the predictive model for clustering and switching. We then applied these predictive models to the resting-state functional connectivity data by computing the resting-state functional connectivity strength for each participant. The model prediction was assessed by the Spearman correlation between the predicted value from the model and the observed values. The CPM-based prediction from the connectivity strength of resting-state functional connectivity was significant for the clustering ($r_s = .48, p < .001$) and switching ($r_s = .21, p = .048$) components. This shows that findings from task-based functional connectivity data largely generalize to independent resting-state functional connectivity data. We believe this new analysis extends the validity evidence of our findings and thank the reviewer for this suggestion.

We included the new analysis of the resting-state functional connectivity analysis in the manuscript. The results were included in the results section "Internal validation: Prediction of PolyFT clustering and switching from resting-state functional connectivity" (p. 17). The materials and methods section was extended accordingly (p. 34), and we address the new finding in the discussion section (p. 21).

R1 Comment #4: *Given the limitations of the sample size with reference to the CPM approach noted in the discussion section, it would be helpful to provide a rationale for why the CPM approach was taken instead of more typical approaches wherein whole brain correlations are computed (e.g., correlation approaches often overfit the data and limit generalizability to novel data; Shen et al., 2017 Nature Protocols).*

We acknowledge that the selection of the CPM approach over more typical approaches was not clearly justified in the methods. We used the CPM with a cross-validation approach since the cross-validation is a more conservative way to infer a brain-behavior relationship than the typical correlation approach (Shen et al., 2017). While the more typical approaches such as correlation or regression models often overfit the data, limiting the generalization of the results to novel data, the cross-validation approach in CPM tests the strength of the brain-behavior relationship in independent data. This allows to increase the likelihood of replication in future studies and preventing the overfitting of the data (Shen et al., 2017).

We now better clarify the selection of our approach and state the advantages of using the CPM approach compared to more typical approaches.

(p. 32-33) “We used CPM with a cross-validation approach since it is a more conservative way to infer a brain–behavior relationship than the typical correlation approach, allowing to increase the likelihood of replication in future studies and preventing the overfitting of the data¹⁰²”.

R1 Comment #5: *In the RJT task, participants had 2 seconds to report their answer. Did this lead to missing data or was no response taken to be a response at the middle of the scale (i.e., the default starting location)? If missing data was an option, how much missing data was there and how was it treated?*

We apologize for the lack of clarity regarding this question. Indeed, after a 2-seconds reflection period, participants had 2 seconds (response period) to move the slider and validate their rating selection within the scale with a left-click. When participants did not manage to validate their responses within the response time, the final position of the slider in the scale at the end of the response time was considered as their relatedness ratings. Please note that in all trials (including unvalidated ones) the slider had been moved by the participants from its initial position. Moreover, on average, participants validated 96.5% of the total number of trials (595 trials). Therefore, in all trials we had a response by the participants that was either validated (96.5% of the trials across participants) or not (3.5% of the trials across participants).

In the materials and methods section entitled “Relatedness judgment task”, we now state this more clearly:

(p. 28) “The final position of the slider in the scale after validation was considered as the semantic relatedness rating. When participants did not validate the trials (3.5% of the trials across participants), the final position of the slider at the end of the response time was considered as the semantic relatedness rating”

To ensure that the number of validated trials did not impact the results of our analyses, we explored the relationships between the number of validated trials and all the metrics we explored for the PolyFT task (clustering and switching), the network metrics (WUN and UUN), creativity tasks (CAT and AUT) and executive function tasks (span, TMT, fluency tasks and Stroop). No significant correlation was observed between the number of unvalidated trials and all outcomes explored.

We now better clarify this issue in the ‘Statistical analyses for the behavioral measures’ part of the materials and methods section:

(p.31): “We ensured that the number of the unvalidated trials during the RJT task was not correlated to any of the PolyFT components, and the measures of creative abilities, SemNet metrics and executive function abilities”.

R1 Comment #6: *Minor points: Unweighted undirected networks were thresholded so that only ratings higher than 50 (middle of the scale) were kept and then binarized. Providing a rationale for this*

threshold would be helpful as would robustness analyses to examine the extent to which associations change depending on the level of the threshold. However, a strength of the current approach is the use of weighted undirected networks that do not require arbitrary thresholding. As such, I don't think additional sensitivity analyses are necessary, a sentence about why a threshold at the middle of the scale is reasonable would suffice.

We agree that the arbitrariness on the thresholding is an important issue in network science. The selection of the threshold for our unweighted semantic networks was based on previous studies by Benedek et al (2017) and Bernard et al (2019) that showed how UUN thresholded at 50/100 predicted creativity. This threshold at the middle of the visual scale appears intuitively plausible as it only keeps associations between concepts that were judged as moderately to highly related. We now better clarify the selection of our threshold in the materials and methods section:

(p. 29): “For the UUN, we selected the threshold value of 50 (at the middle of the visual scale, hence retaining associations between concepts only when rated as moderately to highly related) based on previous studies^{24–25} that showed how UUN metrics thresholded in this way predicted creativity.”

As the reviewer correctly noticed, we also computed the metrics in the weighted networks, and because the issue of the thresholding was also raised by another reviewer, we now provide a new computation of the metrics using a similar approach to the one proposed for the brain networks (Bassett et al., 2012; Fornito et al., 2016). We considered a range of seven thresholds (35 to 65 with an interval of 5) and computed the area under the curve for the different unweighted network metrics. Then, statistical analyses were performed on this unique area under the curve value for each network metric that correlated to the PolyFT component switching. The correlation between the switching was significant for the metrics of *CC* ($r_s = .258, p = .017$), *Q* ($r_s = -.231, p = .033$) and *efficiency* ($r_s = .269, p = .013$). These results suggest that by avoiding the arbitrariness of the threshold definition, our initial results related to UUN remain significant. Importantly, this data-driven process validates the selection of our initial threshold. We added these analyses in the supplementary material.

R1 Comment #7: *Minor points: There are moments where the framing of the purpose of the study does not map appropriately onto the methods. For example, “Assessing functional brain connectivity associated with individual differences in clustering and switching may provide a useful approach to capture the brain systems that underlie distinct memory search components”. I think the language here (and elsewhere) could be tempered to indicate that neural correlates of between-person differences in distinct memory search components are being revealed by the methods.*

We have checked the manuscript to ensure that the framing of our study maps appropriately onto the methods we used. For example, in the introduction, we now state:

(p. 5) “Assessing functional brain connectivity associated with individual differences in clustering and switching may provide a useful approach revealing the neural correlates that underlie individual differences in distinct memory search components”.

Reviewer 2:

Ovando-Tellez and colleagues employed a series of behavioral assessments and a novel fMRI paradigm - relatedness judgment task – to tease out latent processes during creative semantic search. The work is highly innovative and comprehensive, and I thoroughly enjoyed reading the manuscript.

We thank the Reviewer for their positive comments and helpful suggestions.

However, there are a few concerns that weaken the significance of the study, as detailed below.

Major:

R2 Comment #1: *RJT task. Each RJT trial was composed of a two-second thinking period and a two-second response period, after which there was a 0.2-0.7s fixation period. The participants could freely move the cursor on a slider and validate their response with a left click. What if participants failed to respond within the given window? If it did occur, how did the subsequent analysis handle missing trials? Did the task performance drive the prediction? Readers could also benefit from a diagram of the RJT in Fig. 1 since this is the main fMRI task. Besides, have authors looked at the main activation effect of RJT?*

We acknowledge this lack of clarity in the methods. When participants did not manage to select and validate their responses within time allowed, we used the final position of the slider at the end of the response time window as their relatedness ratings. In all trials (including unvalidated ones) the slider had been moved by the participants from its initial position. Moreover, on average, participants validated 96.5% of the total number of trials (595 trials). Therefore, there were no missing trials. Please, see the **R1 Comment #5**.

In the materials and methods section entitled “Relatedness judgment task”, we now state this more clearly:

(p. 28) “The final position of the slider in the scale after validation was considered as the semantic relatedness rating. When participants did not validate the trials (3.5% of the trials across participants), the final position of the slider at the end of the response time was considered as the semantic relatedness rating. Note that in all trials (including unvalidated ones) the slider had been moved by the participants from its initial position.”

We agree that a diagram of the RJT task could help to better understand the task. We revised Figure 1 and added a more complete diagram with the whole trial representation.

Regarding the main activation effect of the RJT, we consider that exploring the brain activation during the RJT as another interesting way to look at the data. However, as exploring the brain

activation associated with judging relatedness between close and remote word pairs is not directly related to aim of the current study, which focuses on clustering and switching during memory search, we did not include them here.

R2 Comment #2: *CPM analysis. In the section “Connectome-based predictive modeling (CPM) of AmbiFlu components,” authors included mean FD in the predictive model to remove the influence of head motion (lines 478 – 480). This is confusing since including mean FD in the predictive model potentially exacerbates the problem if mean FD correlated with AmbiFlu components. If authors want to rule out the head motion, authors can run CPM without mean FD and later conduct a partial correlation analysis between predicted and true AmbiFlu scores controlling for motion.*

We thank the Reviewer for pointing out this issue about the mean FD. As suggested by the reviewer, we reran the CPM analysis without using the mean FD as a regressor in the predictive model. Then, for each CPM analysis we calculated the Spearman partial correlation between the predicted and the real value of the PolyFT components, controlling for the mean FD. The results of the Spearman partial correlations between the predicted and real values of the PolyFT components remained significant for both the clustering ($r_s = .360, p < .001$) and switching ($r_s = .395, p < .001$) components. Since the mean FD does not correlate with any of the clustering ($r_s = .016, p = .89$) and the switching ($r_s = -.051, p = .64$) components, we kept the analysis using the mean FD as a covariate in the predictive model in our manuscript, and additionally report the analyses suggested by the Reviewer in the Supplementary Material as evidence of the robustness of these findings.

R2 Comment #3: *Please also report the negative CPM network. The author wrote, “Note that the negative model networks only included a small number of links.” For the clustering component, the positive network contains 43 edges while the negative network has 14, and I would not call it a small number. The negative network could be used to examine the hypofrontality hypothesis that a loosened top-down regulatory control boosts creativity abilities (Chrysikou et al., 2013). It would be interesting to see if the negative switching model consists of negative links in DAN (or a decoupled left IFG) to enable less constraint exploration (Chrysikou et al., 2013; Kleinmintz et al., 2018; Xie et al., 2021)?*

We thank the Reviewer for raising the idea of discussing the negative network in relation to the hypofrontality hypothesis.

We examined the negative networks to look for arguments for a top-down regulatory control related to increased creative abilities. The negative network predicting *clustering* included interactions between DMN and temporoparietal networks, and within the salience network. The negative network predicting *switching* included interactions between DMN and ECN networks, but no involvement of the DAN and/or salience networks. We did not observe the involvement of DAN and/or left lateral prefrontal nodes in clustering and/or switching negative networks. On the contrary, higher connectivity (positive predictive network) in ECN and DAN were related to higher clustering and higher AUT performance, and thus, not supporting the hypofrontality hypothesis.

We now discussed the negative networks in relation to the hypofrontality hypothesis in the discussion section:

(p. 23) “The negative model networks predicting both clustering and switching do not involve attentional or cognitive control regions such as the DAN or the lateral frontal cortex. On the contrary, higher connectivity in ECN and DAN was related to higher clustering and higher AUT performance. Hence, our results do not support the hypothesis of a release of regulatory control leading to higher creative abilities¹⁴⁰ (see also¹⁴¹).

As suggested by the Reviewer, we now report the negative networks for *clustering* and *switching* components in the Supplementary Material (SIS2), and removed the claim about the size of the negative networks (i.e., “*Note that the negative model networks only included a small number of links*”). In addition, we included two new Supplementary Figures (Figure S2 - S3) to illustrate the negative predictive networks for *clustering* and *switching*.

R2 Comment #4: *Graph analysis. When calculating the semantic network metrics for binary graphs, authors chose an arbitrary threshold of 50 to binarize the weighted semantic graph. This is potentially problematic since using the same threshold across subjects would yield binary graphs with different densities, altering graph metrics. For example, the authors claimed, “These results suggest that individuals with higher switching during the AmbiFlu task had a more efficient SemNet, that was more connected and less modular” (Line 548-550). Could a more connected and less modular network arise simply because the participants rated everything as highly related, leading to a denser graph? Alternatively, instead of using a fixed threshold across subjects, authors could keep 50% of the edges of the semantic network, which would make the graph metrics more comparable across subjects. Also, it would be interesting to run a consensus clustering using consensus_und.m in brain connectivity toolbox on the individual-level communities to obtain a group-level community, just to see if the identified communities correspond to any meaningful categories.*

We acknowledge that using the same threshold across participants rely on different graph densities. Previous studies showing that more creative individuals rate unrelated words as less semantically distant (Rossmann and Fink, 2010) and are faster in judging the relatedness of concepts compared to less creative individuals (Vartanian et al., 2009). So, we considered that selecting the judgment given by the participants from 50 to 100 captured only the moderately to highly related associations to them, while maintaining individual differences in the average level of perceived relatedness which appears to capture relevant creativity-related variance.

As the issue of the density of the UUN graphs was also raised by another Reviewer, we rephrased relevant parts of the result and discussion. Please see also our reply to **R1 Comment #2**.

(p.11): “These results suggest that individuals with higher switching during the PolyFT task had a more efficient SemNet, that was more connected and less modular” has been changed to “The correlation between Eff in the UUN network and the switching component remained

significant after FDR correction. These results suggest that individuals with higher switching during the PolyFT task had a more efficient SemNet”

(p.24): “Fourth, when exploring the UUN graph metrics, the threshold applied to the networks could yield binary graphs with different densities, which may impact the graph metrics. Future studies should address this issue by developing novel approaches to explore SemNet metrics independent of the network density”.

In an attempt to avoid the arbitrariness of the threshold, we explored the UUN metrics using the AUC methods as proposed for fMRI network analyses (Bassett et al., 2012; Fornito et al., 2016; please see the reply to **R1 Comment #6**). We explored the UUN metrics based on the AUC obtained from seven different thresholds, from 35 to 65 with an interval of 5. We explored the correlations between these values and the PolyFT components yielding similar results to those reported in the manuscript. We report this new analysis in the Supplementary Material (SI S1).

Finally, we thank the Reviewer for the suggestion of exploring the consensus clustering. We agree that it is interesting to investigate whether running a consensus clustering on the individual-level communities to obtain a consensus-based group-based community partition and examine whether any meaningful communities appear. We ran this analysis as suggested. However, this analysis did not reveal any meaningful communities at the group-based consensus-based partition. This is likely due to the process we used to select the words for the RJT, which are based on controlling for distance between the words and not on a priori communities (Bernard et al., 2019).

R2 Comment #5: Multiple comparisons. Line 522-524: “The correlations between clustering and both AUT-fluency and AUT-uniqueness remained significant after FDR correction for multiple comparisons.” How many comparisons were corrected for? All the pairwise correlations in Fig. 2? Given how many correlations were computed in Fig. 2 ($20 \times 19 / 2 = 190$), it seems unlikely that all p s < 0.01 passed FDR. Also, it may be clearer if authors just report significant p -values after correction, instead of first reporting raw p -values and then stating what remained significant after correction.

As the Reviewer correctly noticed, we performed an FDR correction for 190 comparisons. The p -values that remained significant after the correction ranged from $p = .007$ to $1.48 \cdot 10^{-28}$.

We reported the p -values for all correlations both before and after the correction for multiple comparisons as a compromise between overly conservative and liberal approaches.

We now clarify this point in the ‘Statistical analyses for the behavioral measures’ part of the materials and methods section:

(p. 31): “We used an FDR approach to correct for multiple comparisons. We reported the p -values for all correlations both before and after the correction for all the comparisons that were performed, as a compromise between overly conservative and liberal approaches”.

R2 Comment #6: *Figures 4 and 5 could be improved. The table in Fig 4(A) and Fig. 5(A) should have “lobe” instead of “network” on top. I don’t see any additional benefits of including panel (A) since all discussion was based on the functional networks. Besides, how was the percentage calculated, and why do they not sum up to 100? Panel B in Fig 4 and 5 is too densely connected and is not very informative. Perhaps show only a few highest-degree nodes. Show the full node-to-node CPM model instead of summed over the network may also help. The degree itself did not mean much without knowing how many nodes each network has.*

We agree that the panel A of Figure 4 and Figure 5 are not further discussed in the manuscript and provides partly redundant information. Therefore, we now removed the Panel A in Figure 4 and Figure 5 from the manuscript. In addition, we modified these figures for a better visualization of the different connections composing the predictive model networks of *clustering* and *switching*. We highlighted the most relevant nodes with respect to their number of connections, and included the total number of nodes for each predictive model network. We believe that these modifications made the figures clearer and more informative. We did not illustrate the full node-to-node CPM model, since it has a high number of connections that may be difficult to visualize (*clustering* = 43 connections; *switching* = 259 connections).

The reported percentage of links reflects how many links in the model network refer to specific functional networks according to the atlas. Each link in the model network connects two different ROIs that belong (or not) to a different functional network. Therefore, the percentage is in relation to the total number of edges in the model network, which can connect nodes within a same network or between different networks. Note that the percentage does not necessarily sum 100%, since each edge ‘belongs’ to one or two functional networks.

To avoid a misunderstanding of the meaning of these percentages, we modified the legend of the Figure 4 and Figure 5 as follows:

(p. 15): *“At the upper right side, a table with the percentage of links connecting nodes that belong to these functional networks is shown. This percentage considers all connections to the regions of a given functional network in relation to the total number of connections in the model network (i.e., 43 links).”*

(p. 16-17): *“At the upper right side, a table with the percentage of links connecting nodes that belong to these functional networks is shown. This percentage considers all connections to the regions of a given functional network in relation to the total number of connections in the model network (i.e., 259 links).”*

Additionally, we provide some examples within the main manuscript:

(p. 14): *“The majority of the links (63% of all connections) connected brain regions belonging to the executive control networks (ECN; in particular the intraparietal sulcus) to diverse regions of other networks, including the salience, visual, and somatomotor networks.”*

(p. 14): “The model network also included links between regions of the dorsal attention network (DAN; in particular the superior parietal lobule) and regions of the visual and somatomotor networks, as well as between the temporal pole (**Figure S1a**) and several regions of the salience network (23 % of all connections).”

(p. 15): A higher number of links connected different brain regions belonging to the DMN to regions of the salience (in particular the frontal operculum), DAN, visual, and somatomotor networks, and to the lateral prefrontal and temporal areas of the ECN (56% of all connections).

R2 Comment #7: *The authors extracted two latent components from the AmbiFlu task, i.e., clustering and switching, and introduced clustering as convergent and deliberate exploitation (more similar to CAT) and switching as a divergent and spontaneous exploration (more similar to AUT). However, the behavioral results did not quite match that framing, as neither switching correlated with AUT nor clustering correlated with CAT performance, which made me wonder about the validity of these latent components. One potential improvement could be the semantic distance across generated words (Beaty and Johnson, 2021; Olson et al., 2021), as some clusters/meanings must be harder to reach than others. Another potential improvement is that instead of using unsupervised data-driven PCA followed by CPM, authors can use methods such as partial least square regression to jointly investigate the behavioral and neural signatures of creativity and then identify the latent behavioral components, for example (Kebets et al., 2019).*

The literature on memory search using fluency tasks indeed interpret clustering as exploitative and switching as explorative. However, it does not necessarily align exploitative with deliberate and switching with spontaneous. On the contrary, Troyer et al.’s work supports another framing in which clustering relates to memory knowledge (more spontaneous/associative memory retrieval) and switching to controlled processes. We did not formulate hypotheses or expectations regarding an alignment of clustering with CAT and switching with AUT, as we believe that CAT and AUT both involve divergent and convergent processes (Gupta et al., 2012; Smith et al., 2013; Volle, 2018). Moving from one cluster to another is necessary to solve CAT trials, because by construction the three cue words are remote from each other, and is thought to involved switching during semantic search (Davelaar et al 2015). In addition, RAT-like tasks correlate strongly with executive tests (Gilhooly & Fioratou, 2009; Lee & Theriault, 2013). Overall, if any, we would rather expect higher links of switching with CAT than with AUT. To clarify our assumptions, we made an extensive revision of the introduction.

We thank the Reviewer’s insightful suggestion of exploring the participants responses using SemDis. However, since there is currently no French SemDis, this analysis would be a whole project in its own, and thus outside the scope of the current manuscript.

Finally, we thank the Reviewer for suggesting the partial least square method which allowed us to explore and become more familiar with this interesting approach. We conducted this analysis on our data and compared it to the CPM approach. Although we found consistency in the results reported in our manuscript by using the partial least square method, we consider that this

approach prevents us to properly address the main aim of this study. In our study, we aimed to first identify the semantic memory search components (clustering and switching) and then characterize them in the context of semantic memory structure, executive abilities, brain connectivity patterns, and creative ability. Using the PCA approach allowed us to identify the components of the PolyFT task, which are theorized to be key components of memory search related to creativity. As the PCA approach confirmed the separability of the clustering and switching components, we then explored how these components related to creative abilities, individual differences in semantic memory structure using SemNets, and executive abilities. In addition, we were able to explore how these components are supported by different brain connectivity patterns. Therefore, although the partial least square is an interesting approach, we consider that by combining the different scores obtained from the PolyFT task and the other behavioral measures in one matrix to maximizes the covariance between this matrix and a brain connectivity related matrix would prevent us to characterize the semantic memory search components separately.

R2 Comment #8: *The organization of the materials and methods section is not easy to follow. The authors conducted multiple behavioral assessments and creative tasks and gave detailed descriptions to all of them in the main text, including many widely-used tasks such as Stroop, AUT, and CAT among others. However, it appeared that the main tasks used in the main analysis were relatedness judgment task (RJT) and ambiguous word fluency (AmbiFlu) task. Thus, the authors could consider only keeping the detailed description of RJT and AmbiFlu in the main text, and briefly introducing other behavioral assessments and moving details to the supplementary materials. Moreover, the “Principal component analysis of the ambiguous word fluency task measures” should be in the Results section. Also, with so many acronyms, the authors should add a glossary of terms, and group them by categories, such as executive functions, network metrics, and creativity.*

We acknowledge that the organization of the material and methods section was not easy to follow. We revised our manuscript based on the Reviewer’s suggestion:

First, we reformatted the manuscript and the result section now comes first in the manuscript. Since the methods section now comes after the results section, we added a short description of the tasks in the results section. We believe that this organization facilitates the understanding of the manuscript.

Second, we moved the “Principal component analysis of the ambiguous word fluency task measures” section to the results section. We only kept the main description of the PCA analyses in the methods section.

Third, to facilitate the reading of the manuscript and the interpretation of the results, we modified the **Figure 1a** by adding the behavioral measurements the different parameters were related to (i.e., executive tests, SemNet metrics and creativity scores). In addition, we added in a new **Table 2** a summary of the creativity tasks, SemNet metrics and executive tests used in the analyses, with the acronym (when appropriate), the different parameters measured by the tasks, and what they measure.

R2 Comment #9: *Minor: Preprocessing does not contain either temporal filtering or detrending. In theory, low-frequency drift or high-frequency cardiac, respiration, and motion-related noise should be rejected in the multi-echo denoising, but considering the data-driven nature of the denoising strategy, I wonder if it is better to remove trends and filter out high-frequency signals before GLM analysis.*

We thank the Reviewer for this comment. Indeed, one of the advantages of using multi-echo denoising and, in particular, TE-dependent analyses (TEDANA) is that the ICA strategy is able to separate noise from BOLD signal. Therefore, as the reviewer noticed, since the multi-echo-ICA denoising removes the non-BOLD components, such as slow drift, motion, cardiac and respiratory, we did not consider applying further methods of denoising.

R2 Comment #10: *Minor: Line 458 should be “pairwise correlations.”*

We corrected this typing error.

R2 Comment #11: *Minor: In the CPM section, the authors stated, “These matrices were Z-Fisher transformed 460 and rescaled by the maximal weight to be in the range of -1 to 1 for the subsequent analyses” (line 460-461). I wonder how the weights are scaled within each connection or participant? Usually, they are z-scored for each connection.*

The pairwise Pearson correlation coefficients computed between the time series of each pair of nodes in the Schaefer atlas were Fisher normalized for each participant. Since we are using a leave-one-out cross validation approach, we did not z-score the data at the group level, to avoid confounding information between the trained (N - 1) and the tested (left out subject) groups.

R2 Comment #12: *Minor: Please report the mean FD across subjects, such as mean FD across subjects (how many subjects with mean FD > 0.5mm) and how many frames were potentially contaminated by head motion (i.e., how many TRs had FD > 1mm). I raised this question because no one was excluded due to excessive head motion, which is not very common.*

We apologize for the lack of precision about this point in the manuscript. To account for the motion artifact, we defined a priori excessive head motion as > 2 mm translation or > 3° rotation. One of the 86 participants in the sample had 4 conspicuous volumes (out of 1996), and two participants had 1 artifacted volume. No run was excessively artifacted to be removed and no participant had a meanFD > 0.5 (mean = 0.106, S.D. = 0.049, Max= 0.37 Min= 0.04). However, we regressed 24 motion parameters during the GLM to avoid the results being contaminated by the motion parameters. We now clarify this point in the materials and methods section:

(p. 32): “No participant was removed due to excessive head motion within a single run, defined a priori as > 2 mm translation or >3° rotation, and no participant had mean FD > 0.5 mm”.

As additional control, we explored the FD between all volumes. We quantified the number of TRs with FD > 1mm. Two of the 86 participants had more than 2% of TRs > 1mm. We ran the CPM analysis after removing these two participants from the dataset (N = 84) and the prediction of both PolyFT components of clustering ($r_s = .368$, $p < .001$) and switching ($r_s = .342$, $p = .002$) remained significant.

R2 Comment #13: *Minor: Add corresponding names for each component in Table 1.*

We added the names of the components instead of the component numbers.

R2 Comment #14: *Minor: Were there any wrong answers in the AmbiFlu task, i.e., answers deemed unrelated to the cue?*

Answers that did not obviously fall into the predefined meaning categories (or that we were not sure of the link with the cue) were classified in a separate category “other”. This category was not considered when scoring cluster size, number of switching, number of meanings. The responses that fall within this “other” category represented < 2% of the total of the responses across participants.

References

- Beaty, R.E., Johnson, D.R., 2021. Automating creativity assessment with SemDis: An open platform for computing semantic distance. *Behav. Res. Methods* 53, 757–780. doi:10.3758/s13428-020-01453-w
- Chryssikou, E.G., Hamilton, R.H., Coslett, H.B., Datta, A., Bikson, M., Thompson-Schill, S.L., 2013. Noninvasive transcranial direct current stimulation over the left prefrontal cortex facilitates cognitive flexibility in tool use. *Cogn. Neurosci.* 4, 81–89. doi:10.1038/jid.2014.371
- Kebets, V., Holmes, A.J., Orban, C., Tang, S., Li, J., Sun, N., Kong, R., Poldrack, R.A., Yeo, B.T.T., 2019. Somatosensory-Motor Dysconnectivity Spans Multiple Transdiagnostic Dimensions of Psychopathology. *Biol. Psychiatry* 86, 779–791. doi:10.1016/j.biopsych.2019.06.013
- Kleinmuntz, O.M., Abecasis, D., Tauber, A., Geva, A., Chistyakov, A. V., Kreinin, I., Klein, E., Shamay-Tsoory, S.G., 2018. Participation of the left inferior frontal gyrus in human originality. *Brain Struct. Funct.* 223, 329–341. doi:10.1007/s00429-017-1500-5
- Olson, J.A., Nahas, J., Chmoulevitch, D., Cropper, S.J., Webb, M.E., 2021. Naming unrelated words predicts creativity. *Proc. Natl. Acad. Sci. U. S. A.* 118, 1–6. doi:10.1073/pnas.2022340118
- Xie, H., Beaty, R.E., Jahanikia, S., Geniesse, C., Sonalkar, N.S., Saggar, M., 2021. Spontaneous and deliberate modes of creativity: Multitask eigen-connectivity analysis captures latent cognitive modes during creative thinking. *Neuroimage* 243, 118531. doi:10.1016/j.neuroimage.2021.118531

Reviewer 3:

This is an interesting and fairly well-written article examining the relationships between semantic memory and creative thinking. The purpose of the research was to examine how switching semantic categories and clustering in the context of a novel ambiguous word association task is related to

creativity in terms of cognitive and neural processing. Among the strengths of the study is the substantial sample size for an fMRI experiment, the individual semantic network metrics, and the novelty of the task design, which is ideally suited to address the research question—as well as being very interesting.

We thank the Reviewer for highlighting the strengths of our study.

I have a few comments on the manuscript, as follows:

R3 Comment #1: *The paper is addressing multiple constructs and at many levels. For example, there is reference to clustering and switching, the relationship with these constructs and associative vs. controlled processing, and then the person-specific neural correlates of these systems. I found the introduction to be somewhat confusing in presenting a clear hypotheses on the anticipated relationships among these constructs and a clear rationale of how understanding these relationships elucidates our understanding of creative thinking. The current introduction seems to move back-and-forth from the one level of analysis to the other, without sufficient and clear framing of the study's specific predictions.*

We apologize for the lack of clarity in the introduction. We carefully revised the introduction along the lines suggested by the reviewer and clarified our hypotheses. All changes are highlighted in the revised manuscript.

We also added the following paragraph in the introduction (p. 6) to summarize our hypotheses and frame our predictions:

“The PolyFT enabled us to assess individual differences in clustering and switching behavior, which are theorized to be key components of memory search related to creativity⁶⁸⁻⁷⁰. We examined how these two components relate to creative abilities, individual differences in semantic memory structure using SemNets²⁴⁻²⁷, and executive abilities⁸⁴. Finally, using a connectome-based predictive modelling (CPM) approach⁸⁸, we explored the functional connectivity patterns that predict clustering- and switching-related components. This approach allowed us to address four hypotheses: First, we expected that both clustering- and switching-related components (as assessed with PolyFT) would correlate with creativity task performance. Such a finding would extend the seminal work from Troyer et al.⁴⁶ to creativity and show that creative thinking involves similar cognitive processes associated with semantic memory search. Second, we hypothesized that switching-related measures would correlate with executive abilities^{46,72,73}. Third, because previous research demonstrated that producing a chain of related words (associates) involves a spontaneous and unconstrained mode of retrieval with little executive demands^{16,18} we expected the clustering-related component of the PolyFT would be related to semantic memory structure as captured by SemNets and more limited executive control. The findings arising from the second and third hypotheses would clarify how the processes framed in different constructs (memory search components, semantic structure, executive functions) relate or differ from each other, and how they correlate with creative abilities. Fourth, we expect that clustering and switching are associated with discriminable brain activation in terms of connectivity patterns between the ECN, DMN, and salience networks.

Characterizing the semantic memory search components of clustering and switching in the context of semantic memory structure, executive abilities, and brain connectivity patterns, and exploring how they relate to creative ability will help to better understand the role of semantic memory search in creative cognition”.

R3 Comment #2: *The connections between the constructs mentioned above and the tasks used in the study add to the confusion mentioned under point #1 above. The links between the constructs and the tasks used to measure them needs to be much better established.*

We carefully revised the manuscript to better explain the links between the constructs and the tasks used. In addition, we added the Table 2 as a summary of the creativity tasks, SemNet metrics and executive tests used in the analyses. We also provide the different parameters we measured and what they measure.

In the introduction we clearly state:

(p. 3): “Semantic memory search depends on the organization of semantic associations stored in memory that drives associative, spontaneous retrieval, and on controlled processes that navigate these retrieval processes based on the context and task demands.” ... “Although associative thinking has proved challenging to measure empirically, studies have linked it to creativity using divergent thinking (generating different new and effective ideas) but also convergent thinking (finding solutions to problems by combining information in novel ways) tasks ^{3,11,14–19}”.

(p. 4-5): ... “The above accounts argue for separable semantic search processes, but little is known about their individual contributions to creative cognition ⁷¹. Broad retrieval ability, the ability to fluently retrieve semantic information from long-term memory, has been reliably associated with creative performance ^{2,39–41}, suggesting that semantic search plays an important role for creative thinking in general.”

(p. 5): “Overall, the clustering and switching components of memory search may both play a role in creativity. Their contribution may vary according to the cognitive processes presumably supporting each component, i.e., memory structure or control processes, respectively. The role of clustering and switching in creativity may also depend on the relative importance of controlled and spontaneous processes in the creativity tasks that are used.”

R3 Comment #3: *In addition to the many levels of analysis, we have—critically—the inclusion of neural data, which yet again adds another layer of conceptual complexity to the project. This is not bad, of course, but it does require a clearer and more streamlined argument in the introduction on how all these levels come together and what they tell us about creativity.*

We revised the introduction to make clear how the different levels investigated (memory processes, executive ability and creative performance, and brain networks) relate to each other and what was hypothesized and expected. See also reply to **R3 Comments #1 and #2**.

R3 Comment #4: *It would be helpful to know if the authors expect that there is a ‘prototypical’ structure for semantic memory that is superseding individual differences.*

This is an important question. We acknowledge that a prototypical semantic memory structure may be underlying but not superseding individual variations of memory structure. Indeed, our results converge with previous work showing individual differences in semantic memory structure (Hills and Kenett, 2022; Kumar 2021). Critically, we add to these previous findings by showing how these individual differences relate to semantic search processes. Future studies may explore how SemNet measures reflect variations/deviations from that prototypical structure.

R3 Comment #5: *What are the study’s predictions? These should be stated clearly and justified by the literature explicitly in the manuscript—as well as how support for one hypothesis over another reveals something important about creativity.*

We thank the Reviewer for raising this point out. We now clearly state our predictions in the introduction and how these hypotheses would reveal important aspects of creative cognition:

(p. 6): “First, we expected that both clustering- and switching-related components (as assessed with PolyFT) would correlate with creativity task performance. Such a finding would extend the seminal work from Troyer et al. ⁴⁶ to creativity and show that creative thinking involves similar cognitive processes associated with semantic memory search. Second, we hypothesized that switching-related measures would correlate with executive abilities ^{46, 72, 73}. Third, because previous research demonstrated that producing a chain of related words (associates) involves a spontaneous and unconstrained mode of retrieval with little executive demands ^{16,18} we expected the clustering-related component of the PolyFT would be related to semantic memory structure as captured by SemNets and more limited executive control. The findings arising from the second and third hypotheses would clarify how the processes framed in different constructs (memory search components, semantic structure, executive functions) relate or differ from each other, and how they correlate with creative abilities. Fourth, we expect that clustering and switching are associated with discriminable brain activation in terms of connectivity patterns between the ECN, DMN, and salience networks.”

R3 Comment #6: *The discussion of the results in many places implies causality. However, the present data—in my view—are unable alone to support any causal claims regarding switching/clustering, controlled/automatic processes, and memory search and structure. Careful reconsideration of some of the language would be strongly recommended.*

We did not mean to imply causality in our discussion. We have checked the manuscript throughout to revised language to avoid potential overstatements.

For example, in the discussion section:

(p. 19): “As expected, the *clustering* and *switching* components differ in their relationships with semantic memory structure and executive control measures, suggesting that they may capture distinct semantic and/or executive processes involved in memory search (see **Figure 1**).”

(p. 19): “Overall, switching in PolyFT seems to rely on both semantic memory structure and executive control processes, which together may support flexibility in memory search.”

(p. 20): “Hence, our clustering component may capture the ability to maximally exploit a given meaning, or the tendency to persist longer in a local/exploitation mode, as opposed to being a global/exploration mode completely free of executive control.”

(p. 20): “These results suggest that *switching* captures a self-paced flexible search behaviour during the PolyFT task that may relies on both semantic memory structure and executive processes.”

R3 Comment #7: *Minor comments: I would encourage the authors to consider re-naming the ambiguous words task—AmbiFlu may not be an optimal choice (especially in a pandemic...)*

We agree that the name of AmbiFlu is not an optimal choice in these times. We replaced the acronym used for our ambiguous polysemous fluency task by PolyFT.

R3 Comment #8: *Minor comments: The plural of corpus is corpora (not corpuses)*

We apologize for this typo. We now corrected it in the manuscript.

References:

- Bassett, D. S., Nelson, B. G., Mueller, B. A., Camchong, J., & Lim, K. O. (2012). Altered resting state complexity in schizophrenia. *NeuroImage*, 59(3), 2196-2207. <https://doi.org/10.1016/j.neuroimage.2011.10.002>
- Benedek, M., Kenett, Y. N., Umdasch, K., Anaki, D., Faust, M., & Neubauer, A. C. (2017). How semantic memory structure and intelligence contribute to creative thought: A network science approach. *Thinking & Reasoning*, 23(2), 158-183. <https://doi.org/10.1080/13546783.2016.1278034>
- Bernard, M., Kenett, Y., Ovando-Tellez, M., Benedek, M., & Volle, E. (2019). *Building Individual Semantic Networks and Exploring their Relationships with Creativity*.
- Cole, M. W., Ito, T., Cocuzza, C., & Sanchez-Romero, R. (2021). The Functional Relevance of Task-State Functional Connectivity. *Journal of Neuroscience*, 41(12), 2684-2702. <https://doi.org/10.1523/JNEUROSCI.1713-20.2021>
- Davelaar, E. J. (2015). Semantic Search in the Remote Associates Test. *Topics in Cognitive Science*, 7(3), 494-512. <https://doi.org/10.1111/tops.12146>

- Fornito, A., Zalesky, A., & Bullmore, E. (2016). *Fundamentals of Brain Network Analysis*. Academic Press.
- Gilhooly, K. J., & Fioratou, E. (2009). Executive functions in insight versus non-insight problem solving : An individual differences approach. *Thinking & Reasoning*, 15(4), 355-376. <https://doi.org/10.1080/13546780903178615>
- Gupta, N., Jang, Y., Mednick, S. C., & Huber, D. E. (2012). The road not taken : Creative solutions require avoidance of high-frequency responses. *Psychological Science*, 23(3), 288-294. <https://doi.org/10.1177/0956797611429710>
- Hills, T. T., & Kenett, Y. N. (2021). Is the Mind a Network? Maps, Vehicles, and Skyhooks in Cognitive Network Science. *Topics in Cognitive Science*, n/a(n/a). <https://doi.org/10.1111/tops.12570>
- Kumar, A. A. (2021). Semantic memory : A review of methods, models, and current challenges. *Psychonomic Bulletin & Review*, 28(1), 40-80. <https://doi.org/10.3758/s13423-020-01792-x>
- Lee, C. S., & Theriault, D. J. (2013). The cognitive underpinnings of creative thought : A latent variable analysis exploring the roles of intelligence and working memory in three creative thinking processes. *Intelligence*, 41(5), 306-320. <https://doi.org/10.1016/j.intell.2013.04.008>
- Rossmann, E., & Fink, A. (2010). Do creative people use shorter associative pathways? *Personality and Individual Differences*, 49(8), 891-895. <https://doi.org/10.1016/j.paid.2010.07.025>
- Shen, X., Finn, E. S., Scheinost, D., Rosenberg, M. D., Chun, M. M., Papademetris, X., & Constable, R. T. (2017). Using connectome-based predictive modeling to predict individual behavior from brain connectivity. *Nature Protocols*, 12(3), 506-518. <https://doi.org/10.1038/nprot.2016.178>
- Smith, K. A., Huber, D. E., & Vul, E. (2013). Multiply-constrained semantic search in the Remote Associates Test. *Cognition*, 128(1), 64-75. <https://doi.org/10.1016/j.cognition.2013.03.001>
- Vanderwal, T., Kelly, C., Eilbott, J., Mayes, L. C., & Castellanos, F. X. (2015). Inscapes : A movie paradigm to improve compliance in functional magnetic resonance imaging. *NeuroImage*, 122, 222-232. <https://doi.org/10.1016/j.neuroimage.2015.07.069>
- Vartanian, O., Martindale, C., & Matthews, J. (2009). Divergent thinking ability is related to faster relatedness judgments. *Psychology of Aesthetics, Creativity, and the Arts*, 3(2), 99-103. <https://doi.org/10.1037/a0013106>
- Volle, E. (2018). Associative and controlled cognition in divergent thinking : Theoretical, experimental, neuroimaging evidence, and new directions. In *The Cambridge handbook of the neuroscience of creativity* (p. 333-360). Cambridge University Press. <https://doi.org/10.1017/9781316556238.020>

REVIEWERS' COMMENTS:

Reviewer #1 (Remarks to the Author):

The authors have been highly responsive to reviewers.

Reviewer #2 (Remarks to the Author):

I appreciated the effort that Dr. Ovando-Tellez and colleagues put into the revision, which has improved significantly. I have only a few minor comments/suggestions.

1. To validate their CPM model, authors used an internal resting-state dataset as the validation set and found significant prediction for clustering ($r_s = .484$, $p < .001$) and switching ($r_s = .214$, $p = .048$) components, showing the robustness of the model. I am curious that what is the authors' thoughts on the higher predictive ability for clustering in the test set ($r = .367$ for training set), since oftentimes the test set would lead to worse performance compared with the training set. Also, as a side note, task-FCs are shown to be similar to rest-FCs (Cole et al., 2014). The reproducibility using same pool of participants may just indicate the high similarity between those two FC patterns.
2. It is interesting that CPM that predicts the Polyfit switching consists of many DMN edges while all of them are between DMN and other networks, and none of them is within-DMN.
3. The connectivity pattern is very crowded, and I suggest that the authors consider using the connectivity visualization tool <https://bioimagesuiteweb.github.io/bisweb-manual/tools/conncontrol.html>.
4. The non-overlapping FC patterns predicting clustering and switching components may be a side-effect of the earlier PCA step to generate these orthogonal components.

Cole, M.W., Bassett, D.S., Power, J.D., Braver, T.S., Petersen, S.E., 2014. Intrinsic and task-evoked network architectures of the human brain. *Neuron* 83, 238–251.
<https://doi.org/10.1016/j.neuron.2014.05.014>

Reviewer #3 (Remarks to the Author):

The authors have addressed most of my concerns from the original round of reviews. The additional analyses provided are also welcome and strengthen the paper.

I would only like to see (perhaps in the discussion) a more explicit statement regarding the current results being reflective of correlational/descriptive relationships that do not address whether these individual differences in network architecture are the cause or the effect of individual differences in creative ability as measured by the tasks used.

To: Editorial Board
Communication Biology

Paris, May 12nd 2022

Please find enclosed a revised version of our manuscript entitled “**An investigation of the cognitive and neural correlates of semantic memory search related to creative ability**”, by Ovando-Tellez et al. We are pleased to read that the editorial board would consider publishing a new version of our paper in *Communication Biology*, conditional to minor revisions. We thank you for the opportunity to revise our manuscript and the reviewers for their comments and new suggestions to improve our manuscript. In the present document, we respond to each comment of the Reviewers in detail, including indications of how we revised the manuscript and excerpts of the revised document. All revisions that have been made are highlighted in the new version of the manuscript to distinguish them clearly. We also checked and edited your manuscript to comply with the format requirements of the journal and to maximised the accessibility.

Reviewer 1 was satisfied with our revised version.

Reviewer 2:

I appreciated the effort that Dr. Ovando-Tellez and colleagues put into the revision, which has improved significantly. I have only a few minor comments/suggestions.

R2 Comment #1: *To validate their CPM model, authors used an internal resting-state dataset as the validation set and found significant prediction for clustering ($r_s = .484$, $p < .001$) and switching ($r_s = .214$, $p = .048$) components, showing the robustness of the model. I am curious that what is the authors' thoughts on the higher predictive ability for clustering in the test set ($r = .367$ for training set), since oftentimes the test set would lead to worse performance compared with the training set. Also, as a side note, task-FCs are shown to be similar to rest-FCs (Cole et al., 2014). The reproducibility using same pool of participants may just indicate the high similarity between those two FC patterns.*

We thank the reviewer for this comment.

Regarding the higher predictive ability for clustering in the test set, since we used the resting data from the same pool of participants, we agree that there may be a high similarity between task-FCs and rest-FCs. We looked at the strength of connectivity of the predictive networks for the clustering and switching components and we observed that for clustering there is a significant correlation between the connectivity during task and rest across participants in both the positive ($r_s = .47$, $p = 6.18 \text{ e-}06$) and the negative ($r_s = .45$, $p = 1.78 \text{ e-}05$) predictive model networks. This correlation for the positive ($r_s = .28$, $p = .008$) and negative ($r_s = .44$, $p = 2.53 \text{ e-}05$) model networks was also significant for the switching component. Hence, we cannot

exclude that part of the results in the resting state data can be explained by this similarity between rest and task functional connectivity.

We added this point as a limitation in the discussion section as follows:

p.18: *“The internal validation using the resting state data of the same participants may be influenced by the similarity between task and rest based functional connectivity data¹⁴².”*

R2 Comment #2: *It is interesting that CPM that predicts the Polyfit switching consists of many DMN edges while all of them are between DMN and other networks, and none of them is within-DMN.*

We agree that this result is interesting. We now pointed this finding in the results section:

p.11: *“No links within DMN network were observed.”*

We also discussed this result in the discussion section:

p. 17: *“Importantly, the connectivity predicting switching involves connections of DMN regions with other networks, in particular with ECN regions and the temporal pole, rather than within-DMN connections, suggesting that switching requires interactions between the DMN and other networks.”*

R2 Comment #3: *The connectivity pattern is very crowded, and I suggest that the authors consider using the connectivity visualization tool <https://bioimagesuiteweb.github.io/bisweb-manual/tools/conncontrol.html>.*

We thank the reviewer for this suggestion. We used the connectivity visualization tool and included one of the views in our figures for a better visualization of the connectivity pattern.

R2 Comment #4: *The non-overlapping FC patterns predicting clustering and switching components may be a side-effect of the earlier PCA step to generate these orthogonal components.*

We agree that it is not surprising to have non-overlapping patterns for the clustering and switching components since the principal component analysis (PCA) identified these two components. However, please note that we used an oblique rotation which does not strictly orthogonalize the components. Nevertheless, we made sure that we do not present the functional connectivity patterns predicting clustering and switching components as non-overlapping patterns, nor did we contrast them directly.

Reviewer 3:

The authors have addressed most of my concerns from the original round of reviews. The additional analyses provided are also welcome and strengthen the paper.

R3 Comment #1: *I would only like to see (perhaps in the discussion) a more explicit statement regarding the current results being reflective of correlational/descriptive relationships that do not address whether these individual differences in network architecture are the cause or the effect of individual differences in creative ability as measured by the tasks used.*

We thank the reviewer for this suggestion.

We added a more explicit statement regarding this point in the discussion:

p. 19: *“Finally, it is important to note that our results reflect brain-behavior relationships that do not address whether the individual differences in brain network architecture are the cause or the effect of individual differences in semantic search components related to creative ability.”*